# Low-level jets in the North and Baltic Seas: Mesoscale Model Sensitivity and Climatology

Bjarke T. Olsen[1], Andrea N. Hahmann[1], Nicolas G. Alonso-de-Linaje[1], Mark Žagar[2], and Martin Dörenkämper[3]

[1]DTU Wind and Energy Systems, Frederiksborgvej 399, 4000 Roskilde, Denmark
[2]Vestas Wind Systems A/S, Aarhus, Denmark
[3]Fraunhofer Institute for Wind Energy Systems, Oldenburg, Germany

**Correspondence:** Bjarke T. Olsen (btol@dtu.dk)

**Abstract.** Low-level jets (LLJs) are wind speed maxima in the lower part of the atmospheric boundary layer. Accurately accounting for these mesoscale phenomena in wind resource assessment is increasingly important as the height of wind turbines continues to grow. During LLJ events, wind speeds increase, leading to a general increase in power output. We utilize wind measurements from LiDARs and a mast for five sites in the North and Baltic Seas to assess the quality of the WRF model simulations for LLJ characterization. We also investigate the benefits of using the WRF model simulations compared to the widely used ERA5 reanalysis. In the WRF model simulations, we vary the grid spacing, the number of vertical levels, and the planetary boundary layer and land surface schemes, modeling configuration choices we deemed most likely to have a substantial impact on LLJ development and morphology. The model's performance is evaluated based on its ability to replicate observed distributions of LLJs and relevant associated characteristics, such as the shear and veer across the rotor-plane of typical large offshore wind turbines (30 m to 300 m). Finally, we generate a five-year LLJ climatology based on using the best-evaluated model configuration. The modeling results show a strong dependency of the LLJ representation and the associated wind profiles on the WRF model configuration and that relying on ERA5 for LLJ characterization is insufficient. For example, the LLJ rate-of-occurrence varied by up to a factor of three or more between some WRF model runs. The simulation using the optimized model configuration more accurately reflects the frequency, intensity, and vertical extension of LLJs, as confirmed by LiDAR data. In the North and Baltic Seas, LLJs occur along the western sea basins around 10–15% of the time, with average jet heights between 140–220 meters, which are well within the height of operation of modern wind turbines. The most LLJ-prone region is east of southern Sweden, especially during spring and summer. These mesoscale phenomena contribute to up to 30% of the wind capacity in some areas in this season. Analysis of the five-year modeled LLJ climatology in the North and Southern Baltic Seas gives insights into the physical mechanisms that create them.

## 1 Introduction

Low-level jets (LLJs) are wind speed maxima in the lower part of the atmospheric boundary layer (ABL). When they occur on a large-scale, they play a vital role in heat, moisture and momentum transport and deep convection, and thus important for the simulation of regional and global climate (Stensrud, 1996; Rife et al., 2010). They are also sometimes responsible

for the transport of pollutants outside urban areas (Darby et al., 2006; Haikin and Castelli, 2022). The formation of LLJs is associated with frictional decoupling (Blackadar, 1957), low-level baroclinicity due to horizontal temperature gradients, large-scale baroclinic zones in sloping terrain (Holton, 1967), and orographic blockage. The first two mechanisms, frictional decoupling, and low-level baroclinicity, typically occur at lower heights (Luiz and Fiedler, 2024) and are relevant for coastal jet formation. Due to the different driving mechanisms, LLJs happen at many spatial and temporal scales. The term "low-level jet" has been used for many of them. Herein, we focus on the jets that form in the lowest part of the atmosphere relevant to wind energy applications (the lowest 500 m) and use the term "low-level jet" in that context.

It is crucial to accurately consider LLJs in wind resource assessment, especially as wind turbines continue to grow taller and encounter a wider range of them. During LLJ events, wind speeds increase, leading to higher power output (Smedman et al., 1996; Gadde and Stevens, 2021). However, the vertical wind shear and veer associated with LLJs can impact turbine performance and reliability (Gutierrez et al., 2017, 2019; Porté-Agel et al., 2020; Gadde and Stevens, 2021; Jong et al., 2024). Due to the increased shear, LLJs may also modify wake dissipation in large offshore wind farms, depending on the height of the LLJ relative to the wind turbine rotor (Gadde and Stevens, 2021).

Many have investigated LLJ characteristics in the North Sea (Kalverla et al., 2019; Wagner et al., 2019) and the Baltic Sea (Smedman et al., 1996; Gottschall et al., 2018; Hallgren et al., 2020, 2022; Rubio et al., 2022), where LLJs are particularly prevalent in spring and early summer when air-sea temperature differences can easily reach 15 °C to 20 °C (Smedman et al., 1996; Hallgren et al., 2020; Rubio et al., 2022). The occurrence of LLJs significantly enhances the available wind resources in these areas. In the Baltic Sea, Smedman et al. (1996) identified frictional decoupling as a key formation mechanism with warm air advecting over cold water, creating a stable marine ABL and an inertial oscillation in space resulting in a super-geostrophic jet. This is akin to the Blackadar mechanism (Blackadar, 1957), except the evolution happens in space, not just in time. Low-level baroclinicity also plays a role. Smedman et al. (1997) shows how the initially stable ABL transitions to a near-neutral and well-mixed (capped) layer as the traveling time over the cold water increases, giving rise to a jet near the capping inversion after this transition.

Because LLJs are transient atmospheric phenomena with a strong diurnal cycle that can be confined to small regions (Stensrud, 1996), they are hard to detect in conventional observing systems, such as SYNOP stations, weather balloons, and satellite remote sensing, which often lack information in the boundary layer. Thus, LLJ assessments have often been done using output from model simulations. In the past decade, the occurrence of LLJs has been verified in various models: ERA5 (Kalverla et al., 2019; Hallgren et al., 2023a; Luiz and Fiedler, 2024), the Weather Research and Forecasting (WRF) model (Rijo et al., 2018; Wagner et al., 2019; Rubio et al., 2022; Aird et al., 2021; Sheridan et al., 2024), and used in case studies (Nunalee and Basu, 2014; Redfern et al., 2023). However, no study to date has produced a high-resolution climatology of LLJs based on an objective and evaluated choice of model parameters, including physical parameterizations.

This study aims to create the first validated climatology of LLJ characteristics for the Baltic and North Seas to support offshore wind energy development. To generate the climatology, we will use the WRF model, by first carrying out a comprehensive sensitivity study and model evaluation, using LiDAR measurements from offshore floating LiDAR systems (FLSs) and one tall mast to choose the optimal model configuration. To generate the LLJ climatology, we run a long-term hindcast simulation

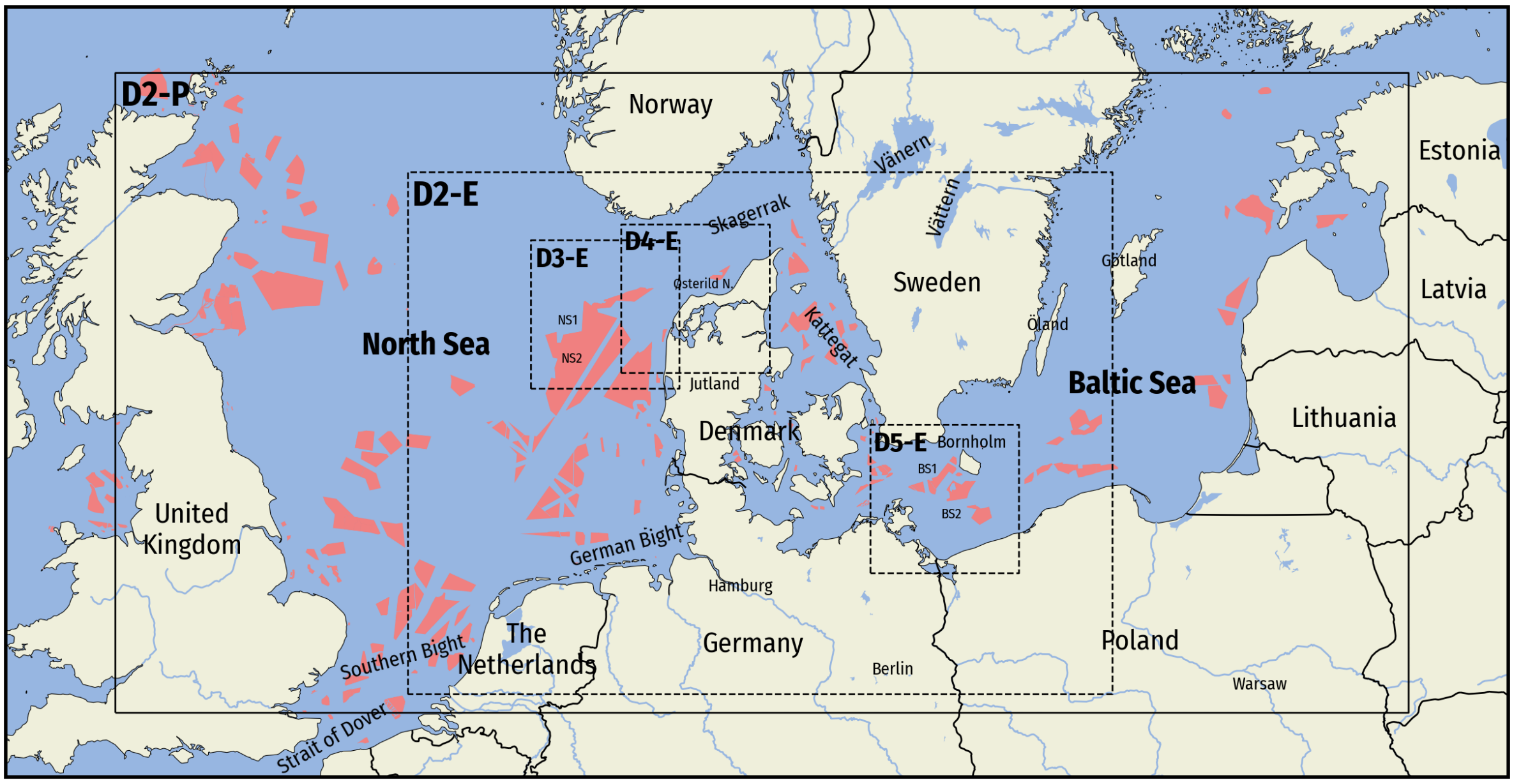

**Figure 1.** Map of the wider study area covering the North and Baltic Seas. The map highlights important bodies of water, islands, and other landmarks referenced in the study. The five observation sites are marked with black crosses and dashed boxes outline the WRF model domains. The outermost WRF domain (D2-P; full black line) is used for the LLJ climatology and the four innermost domains (D2-E – D5-E; dashed black lines) are used during model evaluation, see Sec. 2.2.3. The D2-E and D2-P domains share a common parent domain (D1; not shown). The coral-colored areas mark current wind farms, farms under construction, or future wind farm development zones from the openly available EMODnet Human Activities database (version dated 2024-05-08).

using the best-performing model configuration. Due to the sensitivity of LLJ rates to classification methods, temporal-spatial resolution, and vertical levels (Kalverla et al., 2019), the climatology will be presented with an emphasis on relative spatial patterns over absolute rates.

The study is organized as follows: Section 2 covers the methodology, including measurements, models, datasets, LLJ criteria, reference turbine, and evaluation metrics. Section 3 details the LLJ characteristics from measured data. Section 4 evaluates the models' ability to capture observed LLJ characteristics. Section 5 presents the climatology results from the long-term WRF model simulation. Finally, in Section 6 and 7, the discussion and the conclusions are provided.

## 2 Methods

The area of focus in this study is shown in Fig. 1. The map also shows the five observation sites, the WRF model domains we used, and important bodies of water, islands, and cities we refer to in the text. The coral-colored areas show where current wind farms, wind farms under construction, and future wind farm development zones are located.

**Table 1.** Measurement instruments, locations, vertical levels, and data availability after filtering and resampling for the five measurement sites

| Site | Instrument | Latitude | Longitude | Levels | Data avail. |
|---|---|---|---|---|---|
| NS1 | ZephIR ZX300 Lidar | 56.6279°N | 6.3019°E | 30, 40, 60, 90, 100, 120, 150, 180, 200, 240, and 270 m | 85 % |
| NS2 | | 56.3444°N | 6.4574°E | | 93 % |
| BS1 | | 54.9944°N | 14.3547°E | | 85 % |
| BS2 | | 54.7170°N | 14.5882°E | | 92 % |
| Østerild N. | Boom-mounted cup anemometers on lightning mast | 57.0870°N | 8.8807°E | 40, 70, 106, 140, 210, and 244 m | 80 % |

### 2.1 Observations

This study considers measurements from five sites: two offshore FLSs in the North Sea (NS1 and NS2), two FLSs in the Baltic Sea (BS1 and BS2), and the Østerild mast (Peña, 2019) in Northern Jutland, summarized in Table 1. The four FLSs are instrumented with ZephIR ZX300 vertical profiling LiDARs sampling at 11 vertical levels every 17.4 s, aggregated to 10 min averages. For comparison with the WRF model and ERA5, the samples were further resampled to 1 h averages. The FLSs measurement periods cover 15 November 2021 to 15 December 2022. The Northern lightning mast at Østerild is equipped with cup anemometers mounted on northward-facing 0° ± 1° booms at 6 heights (excluding one at 7 m; see Table 1). Following Peña (2019), measurements from 133° to 192° wind direction are excluded to avoid wind shadow effects from the test turbines at the site. All five sites were assessed over the same period: 15 December 2021 to 14 December 2022 (8760 h total).

The FLS data underwent quality control and assessment, including filtering via status flags from the sensor, flagging, and removal of unrealistic values and duplicates. Observations with missing data at any height were further filtered for vertical consistency. The 10-minute averages were aggregated to 1-hour averages (when at least 50 % of samples are available) for consistency with ERA5 and WRF model output. After filtering, between 80 % (7003 at Østerild N.) and 93 % (8132 at NS2) of the 8760 h samples remain, see Table 1.

### 2.2 Models

#### 2.2.1 The ERA5 reanalysis

The ERA5 reanalysis (Hersbach et al., 2020) is a global gridded reanalysis product that represents the best estimates of weather conditions that span from 1940 to the present day. It consists of hourly variables on 137 model levels in 0.25° × 0.25° grid cells forming a regular global grid. In the study, it serves as a well-established baseline, commonly used in wind energy applications. It has also been extensively used in studies related to LLJs (Kalverla et al., 2019; Rubio et al., 2022; Sheridan et al., 2024;

Hallgren et al., 2023b; Luiz and Fiedler, 2024). Herein, we use the bottom 16 levels (index 122–137) which gives us levels that span above 500 m at our particular sites.

#### 2.2.2 The NEWA wind atlas

The New European Wind Atlas (NEWA) data set (Dörenkämper et al., 2020) is also used as a baseline comparison with the model simulations. Wind speed and other meteorological parameters are available at 50, 75, 100, 150, 200, 300 and 500m above ground with a horizontal grid spacing of 3 km. In the NEWA project best setup for the model was identified by a large set of simulations and compared in terms of the accuracy of the simulated wind speed distribution (Hahmann et al., 2020), but not concerning its depiction of LLJs. The NEWA WRF model simulations used version 3.8.1 and were extended to cover the required time period in this study. Given the limited number of vertical levels available from NEWA, we only include it in parts of the model evaluation focused on spatial variations in LLJ occurrence.

#### 2.2.3 The WRF model simulations

We used the Advanced Research WRF (WRF-ARW) model (Skamarock et al., 2019) to derive a spatially consistent high-resolution climatology of LLJs in the North and Baltic Seas. The modeling work is divided into three phases labeled “E”, “S”, and “P” (experiment, sensitivity, and production) in Table 2. First, we created the E runs by performing an ensemble of WRF model simulations to identify the model configuration that best simulates the wind climate and the occurrence and characteristic of LLJs for a set of cases, as explained later in section 4.1. The E simulations are short, covering 36 hours, where the first 12 hours are ignored (spin-up). The configuration of the WRF model domains for the E set of runs (dashed lines in Fig. 1) is as follows. We used an outer domain (D1) with 9 km grid spacing and a smaller domain (D2-E) of 3 km grid spacing. Within E2, there are three smaller localized domains (D3-E – D5-E) with a 1 km grid spacing centered within each measurement site.

Secondly, we ran a smaller set of WRF model simulations covering a full year to identify whether the method to initialize and the duration of the model simulations were significant to the simulation of LLJs. This second set of simulations, the “S” runs in Table 2, was run with the best-performing model identified in the E set. We used the same domains for the S simulations as we did for E.

Lastly, we conducted a multiyear simulation spanning the period 26 June 2019 to 26 June 2024 for a larger geographic area covering the North and Southern Baltic Seas (Fig. 1). This multi-year simulation is labeled “P_3DTKE”. For this “climatology” simulation, we used the same 9 km outer domain as for E and S, but for the nested 3 km domain D2-P (outer full lines in Fig. 1), we extended the domain to the west and east and slightly north to cover the North Sea and the southern Baltic Sea. The 1 km inner domains were not used for the production of the climatology.

It is worth mentioning that two modifications were made to the land use determination. First, several large lakes were transformed to sea, since their temperatures are included in the OSTIA dataset (Table 3). Second, much of the coastal area of the Wadden Sea from Holland to Denmark is converted from “swamp” to “tidal zone” as explained in (Hahmann et al., 2020).

**Table 2.** Naming and description of the ensemble of the WRF model experiments. Reference is given when first cited.

| Name | PBL scheme | SL scheme | Levels | Additional changes |
|---|---|---|---|---|
| E_3DTKE | 3D TKE (Zhang et al., 2018) | MM5 similarity (Jimenez et al., 2012) | 85 | |
| E_3DTKE_NUD | 3D TKE | MM5 similarity | 85 | Spectral nudging on D1 |
| E_BL | BouLac PBL (Bougeault and Lacarrere, 1989) | MM5 similarity | 85 | |
| E_MYJ | MYJ (Janjić, 1994) | Eta similarity (Janjic and Zavisa, 1994) | 85 | |
| E_MYNN_ML0 | Mellor-Yamada Nakanishi and Niino (MYNN) Level 2.5 (Nakanishi and Niino, 2009) | MYNN SL (Nakanishi and Niino, 2009) | 85 | bl_mynn_mixlength = 0 |
| E_MYNN_L125 | MYNN | MYNN SL | 125 | bl_mynn_mixlength = 2 (Olson et al., 2016) |
| E_MYNN_L55 | MYNN | MYNN SL | 55 | bl_mynn_mixlength = 2 |
| E_MYNN_NUD | MYNN | MYNN SL | 85 | bl_mynn_mixlength = 2; spectral nudging |
| E_YSU | Yonsei University scheme (Hong et al., 2006) | MM5 similarity | 85 | |
| S_3DTKE_3D | 3D TKE | MM5 similarity | 85 | Spectral nudging in D1, 3 days runs, 12 h spinup |
| S_3DTKE_7D | 3D TKE | MM5 similarity | 85 | Spectral nudging in D1, 7 days runs, 24 h spinup |
| P_3DTKE | 3D TKE | MM5 similarity | 85 | Spectral nudging in D1, 7 days runs, 12 h spinup, climatology domain |

**Table 3.** WRF model configuration used in all the simulations

| Parameter | Option |
|---|---|
| WRF model version | 4.2.1 |
| Grid spacing ($\Delta x, \Delta y$)* | E domains: 5 one-way nested domains 9 km (D1), 3 km (D2-E), 1 km (D3-E, D4-E, D5-E) |
| | P domains: 2 one-way nested domains 9 km (D1), 3 km (D2-P) |
| | The same outer domain (D1) is used for all runs. E domains were also used for the S runs. |
| Time step | Adaptive |
| Terrain data | Global Multi-resolution Terrain Elevation Data 30 " (Danielson and Gesch, 2011) |
| Land use data | CORINE 100m (Copernicus Land Monitoring Service, 2019), ESA CCI (Poulter et al., 2015) where CORINE not available |
| Dynamical forcing | ERA5 reanalysis (0.25°× 0.25°) on pressure levels (Hersbach et al., 2020) |
| Sea conditions | OSTIA SST and sea-ice (0.05°× 0.05°) (Donlon et al., 2012) |
| Land surface model | NOAH LSM (Tewari et al., 2004) |
| Microphysics | WSM5 (Hong et al., 2004) |
| Radiation | RRTMG, 12 min call frequency (Iacono et al., 2008) |
| Cumulus | Kain-Fritsch scheme in D1 (Kain, 2004) |
| Diffusion | diff_opt=2 evaluates mixing terms in physical space (stress form) by applying a turbulence parameterization based on the Smagorinsky first-order closure. |
| Advection | Positive definite advection of moisture and scalars |

Table 3 shows the settings used in all WRF model simulations. For the sensitivity experiments we use five PBL parameterizations: 3DTKE (Zhang et al., 2018), BouLac (Bougeault and Lacarrere, 1989), MYJ (Janjić, 1994), MYNN level 2.5 (Nakanishi and Niino, 2009) and YSU (Hong et al., 2006), and three surface layer (SL) schemes: MM5 similarity (Jimenez et al., 2012), Eta similarity (Janjic and Zavisa, 1994) and the default SL used with the MYNN PBL scheme (Nakanishi and Niino, 2009). To limit computational demand, we chose a sparse ensemble matrix and only varied parameters for one of the PBL schemes, MYNN. For this scheme, we changed the number of vertical levels (55, 85, or 125) and included spectral nudging for one member in the outer model domain (D1). Because the NEWA simulations use an earlier version of the MYNN scheme with a default bl_mynn_mixlength = 0, this configuration was also included in the initial ensemble.

## 2.3 Reference turbine

To illustrate the impact and relevance of a specific offshore wind turbine, we use the IEA 15 MW turbine (Gaertner et al., 2020) as a reference. This turbine model has a proposed hub height of 150 m and a rotor diameter of 242 m, spanning heights from 29 m to 271 m, nearly matching the FLSs scan levels (30 m to 270 m). The turbine has a cut-in and cut-out wind speed of 3 m $s^{-1}$ and 25 m $s^{-1}$, while rated power is reached at 10.59 m $s^{-1}$.

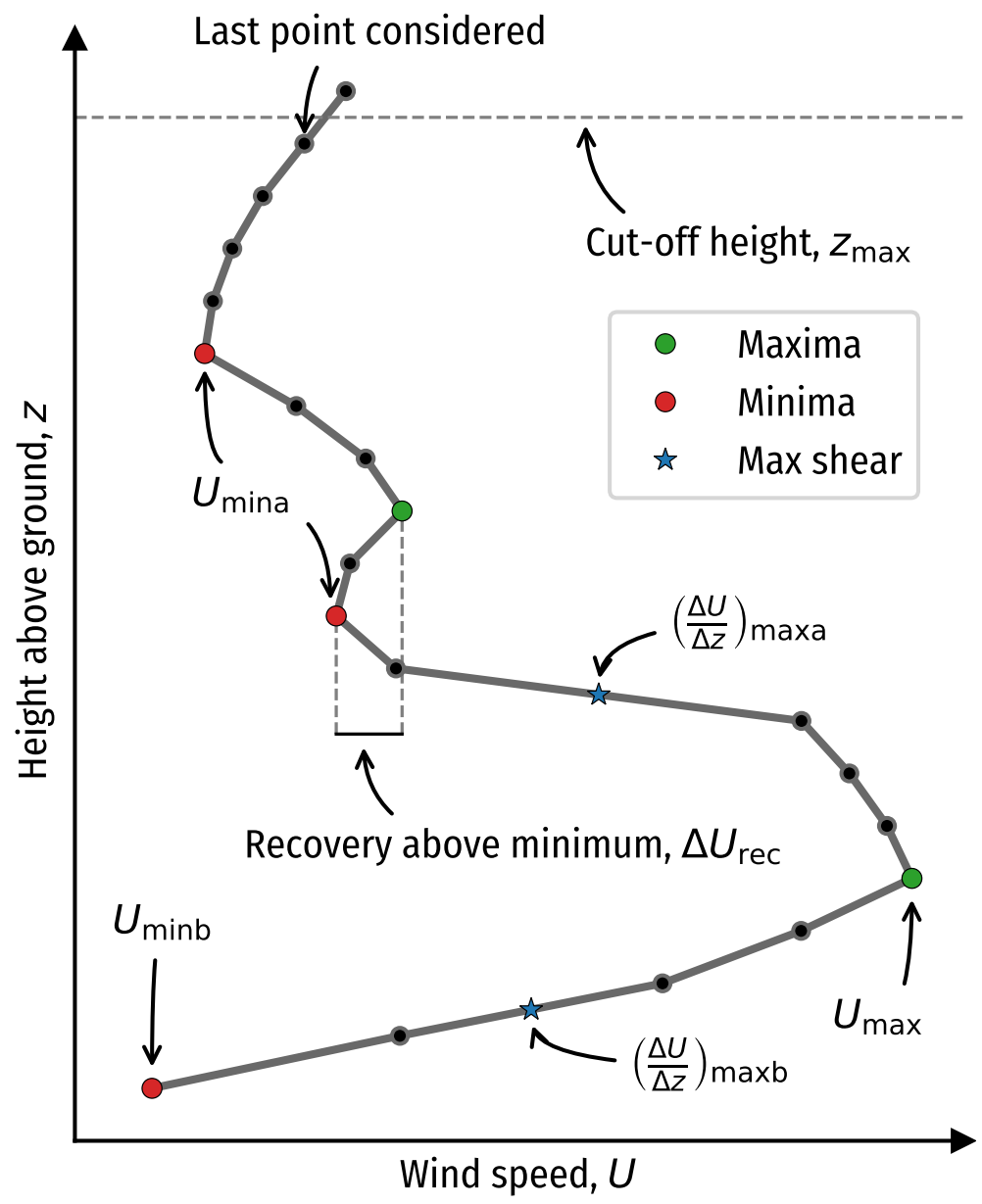

**Figure 2.** Schematic of a wind speed profile and the various parameters used to detect the occurrence of a LLJ.

### 2.4 Low-level jet detection

Classifying whether a vertical wind speed profile is a LLJ event or not involves two steps: (1) identifying reference levels on the vertical profile for jet metrics, and (2) filtering out undesirable profiles based on specific criteria calculated from the reference levels. The profiles may be truncated at a certain height to focus on a specific atmospheric layer.

First, we identify a local maximum in wind speed (i.e. the "jet maximum"). Following Baas et al. (2009), we consider only the lowest 500 m of the atmosphere. When multiple minima are present above or below the jet maximum, a 1 m s$^{-1}$ wind speed recovery is required for a local minimum to be accepted, as illustrated in Fig. 2. Profiles without a local maximum are classified as non-LLJ events. Potential LLJ profiles are filtered using the absolute $(U_{\mathrm{max}} - U_{\mathrm{min}})$ and relative $(U_{\mathrm{max}} - U_{\mathrm{min}})/U_{\mathrm{max}}$ wind speed falloffs. To obtain sufficient samples, we use thresholds of 1.5 m s$^{-1}$ and 15 % falloff, instead of 2 m s$^{-1}$ and 20 %. Different thresholds are applied in other specific situations as detailed in the text.

There is no strong consensus on the most appropriate LLJ detection criteria in the literature, and it will, to some extent, be specific to the context. Hallgren et al. (2023b) suggests a shear-based definition for wind energy applications, as it captures sharp wind profile transitions and is less sensitive to the vertical window. The detection of a LLJ is influenced by the chosen definition, spatio-temporal variability, and dataset resolution (see e.g. Kalverla et al. (2019)). Using 10-min averages from measurements results in more LLJs than longer averaging periods and mesoscale data. High sample density is crucial to accurately resolve the LLJ structure, as they can be short-lived and shallow. Therefore, extra care is needed when comparing studies. Comparing the absolute occurrence rate of LLJs is problematic unless vertical levels, data sampling, averaging, and detection criteria are consistent.

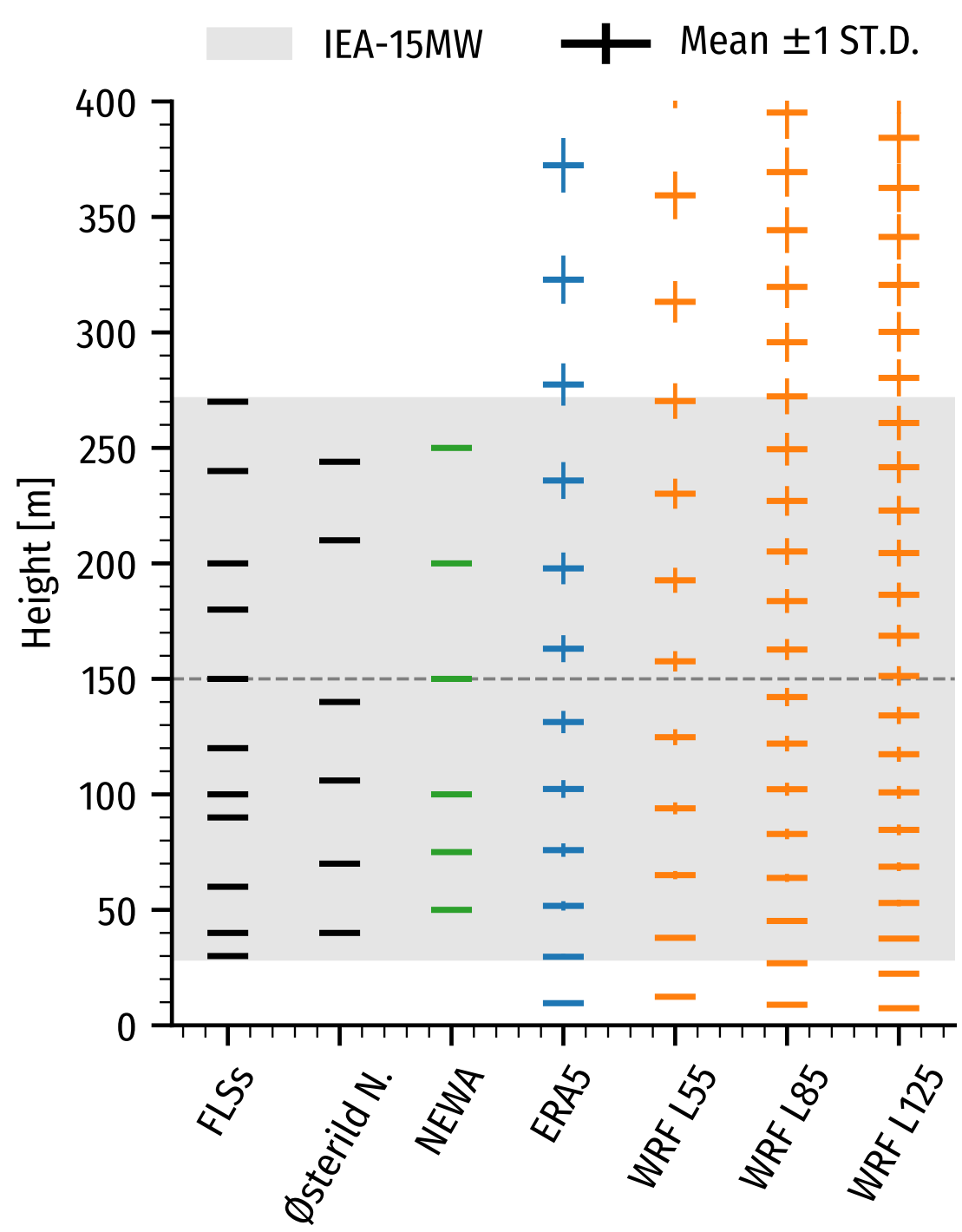

**Figure 3.** Height of the wind speed samples from the floating LiDAR measurements (FLSs), the sonic anemometers on the Østerild mast, the levels available from NEWA, and the model levels ± one standard deviation for ERA5 and the three WRF model configurations, using 55, 85, and 125 vertical levels. The IEA 15 MW rotor-swept heights are shown in grey.

## 2.5 Vertical levels

The FLS-based measurements are aggregated to fixed height levels listed in Table 1, alongside the Østerild mast measurement heights. The ERA5 and WRF models use time-varying vertical levels, shown in Fig. 3, with average heights (horizontal line) and ±1 standard deviation (vertical line). The span of the IEA 15 MW wind turbine is depicted in gray. The number of levels within the rotor plane is 11 for FLSs, 6 for Østerild mast, 8–9 for ERA5, 7–8 for the WRF-L55, 11–12 for the WRF-L85, and 14 for WRF-L125 (Table 2).

For direct comparison between measurements and model simulations (Section 4), model data are interpolated to measurement heights using log-linear interpolation of wind speed, while wind directions are obtained by linear interpolation of the wind components $U$ and $V$. For the final WRF-modeled long-term LLJ climatology, we use the native WRF model levels with variable heights for LLJ identification, restricted to the lowest 500 m of the atmosphere.

### 2.6 Evaluation metrics and LLJ characterization

To assess model performance, we use metrics relevant to LLJ characterization, wind resource assessment, and wind power modeling. Due to time lags in simulated atmospheric features, the modeling accuracy for individual LLJ events is often limited. Therefore, we focus on modeling the distribution of LLJ characteristics over the 70-day evaluation period, rather than individual events. Additionally, to ensure the models perform well in all weather conditions, we include traditional wind power metrics, reflecting state-of-the-art NWP-hindcasts.

Rotor-equivalent wind speed (REWS) (Wagner et al., 2014) and its conversion to power via the reference power curve are used to assess wind speed and power production. The REWS accounts for the rotor-area averaged wind speed and veer, providing a realistic impact on turbine conditions and reducing sensitivity to evaluation level choice.

As a statistical distance measure, we use the Earth Movers Distance (EMD, Rubner et al., 1998), originally proposed by (Kantorovich, 1960). The EMD measures the statistical distances between observed and modeled distributions. The EMD quantifies the "work" needed to align distributions, corresponding to the area between marginal cumulative distribution functions (CDFs) for 1D distributions, capturing both overlap discrepancies and mean distance between these discrepancies.

The following metrics are used in the model evaluation:

– To evaluate power hindcast accuracy, we use the REWS root-mean-square error (RMSE) and Pearson correlation coefficient ($R^2$).

– To evaluate the mean annual energy production accuracy, we use the mean-percentage error (MPE), MAPE, and the statistical distance between empirical distributions for both REWS and hub-height wind direction. The accuracy of the wind power climatology is measured using the EMD for REWS and hub-height wind direction at 150 m, denoted $D_{150}$.

– To evaluate the accuracy of the wind shear ($\Delta U/\Delta z$) and veer ($\Delta D/\Delta z$) across the rotor plane we use the EMD between modeled and observed distributions for all levels in the $m$ area (30 m to 270 m), averaged equally with height. These vertically-averaged metrics are denoted MEMDS (shear) and MEMDV (veer) and favor models that replicate the correct distribution at each height, unlike simple average measures.

– To evaluate the LLJ characterization, we focus on mean rate-of-occurrence and distribution accuracy, using MAPE, MPE, and EMD of $D_{150}$ during LLJ events. The EMD is also used for hourly and monthly LLJ rates, core heights, and wind speeds to assess the diurnal and annual cycle and the core height and speed characteristics of the LLJs. The MEMDS and MEMDV during LLJ events evaluate the shear and veer distributions under these conditions.

– To compare the normalized spatial variability between ensemble members or models we use Z-scores, or "standard scores", of the LLJ occurrence rates. The Z-score is a statistical measure that describes a value's relationship to the mean of a set of values, expressed in terms of standard deviations from the mean. It is calculated using the formula

$$Z = \frac{(X-\mu)}{\sigma}$$

where $X$ is the value being measured, $\mu$ is the mean of the dataset, and $\sigma$ is the standard deviation. Here, the samples are the individual LLJ occurrence rates in each model grid cell. This measure is useful for standardizing different datasets, allowing for comparison across different scales. A high positive Z-score indicates the value is significantly above the mean, while a high negative Z-score indicates it is significantly below the mean. Z-scores are widely used in various fields to identify outliers and to normalize data for further statistical analysis.

– To indicate the best-performing ensemble member, We use the equal-weighted rank of scores. The score simply ranks the models according to the scores from 1 to $n$, with the best-performing model getting rank 1, the second-best rank 2, and so on. The ranks are then averaged across all scores with an equal weight. Lower values thus indicate a “better performing" model. The score provides an objective ranking of model performance but should not be seen as the definitive answer of which model is “best”, a further detailed analysis of each metric and other factors is needed for that.

We conduct the data analysis using Python. The `wasserstein_distance` function from the SciPy package (Virtanen et al., 2020) is used to calculate EMD for linear data, while the `wasserstein_circle` function from the POT (Python Optimal Transport) package (Flamary et al., 2021) is used for circular data (e.g., wind direction, hour of the day, month of the year). All maps are made using matplotlib (Hunter, 2007) and Cartopy (Met Office, 2010 - 2015) with basemaps from https://www.naturalearthdata.com/downloads/.

We also present the duration of LLJ events. To calculate this duration, LLJ occurrence times are grouped into events by counting consecutive LLJs in each time series. A one-time-stamp gap is allowed, with longer gaps indicating separate events.

## 3 Low-level jet observations in the evaluation period

Figure 4 shows the time-height evolution of the wind speed for the entire evaluation period, with detected LLJ cores marked as red dots and WRF model ensemble simulation days as black dots. The figure highlights the missing data periods, particularly in May for NS1, parts of March and December 2022 for BS1, and south-easterlies filtered out from the Østerild N. measurements. LLJs occur more often in the Baltic Sea sites with 138 (≈1.86 %) cases at BS1 and 173 (≈2.14 %) at BS2, respectively. In the North Sea area, the occurrence is below 1 % in all three sites.

Figure 5 presents the seasonal and diurnal LLJ detection rates (a and b), and the distributions of LLJ duration and core heights (c and d). Seasonality greatly influences the LLJ rates, with spring and summer having the highest occurrences. For instance, for BS1 and BS2, the spring and summer rates are 2 % to 4 % compared to less than 1 % during the rest of the year. In the winter months, January and February, almost no LLJs are detected. The BS1 and BS2 sites exhibit a fairly strong diurnal cycle, with about 40 % to 50 % more LLJs in the late afternoon and evening compared to the average rate, while the NS1 and NS2 show limited diurnal variation, except for a morning spike at NS1. Østerild N. favors night-time LLJs and has fewer LLJs around mid-day. Most LLJs last one hour or less, with over 50 % appearing in only one one-hour sample. Events lasting longer than two to three hours are rare. BS1 and BS2 have a bit more long-lasting LLJs than the rest. LLJ core heights mostly range from 45 m to 165 m, while the averages are about ≈106 m for BS1 and BS2, ≈104 m at NS1, ≈121 m at NS2, and ≈117 m at

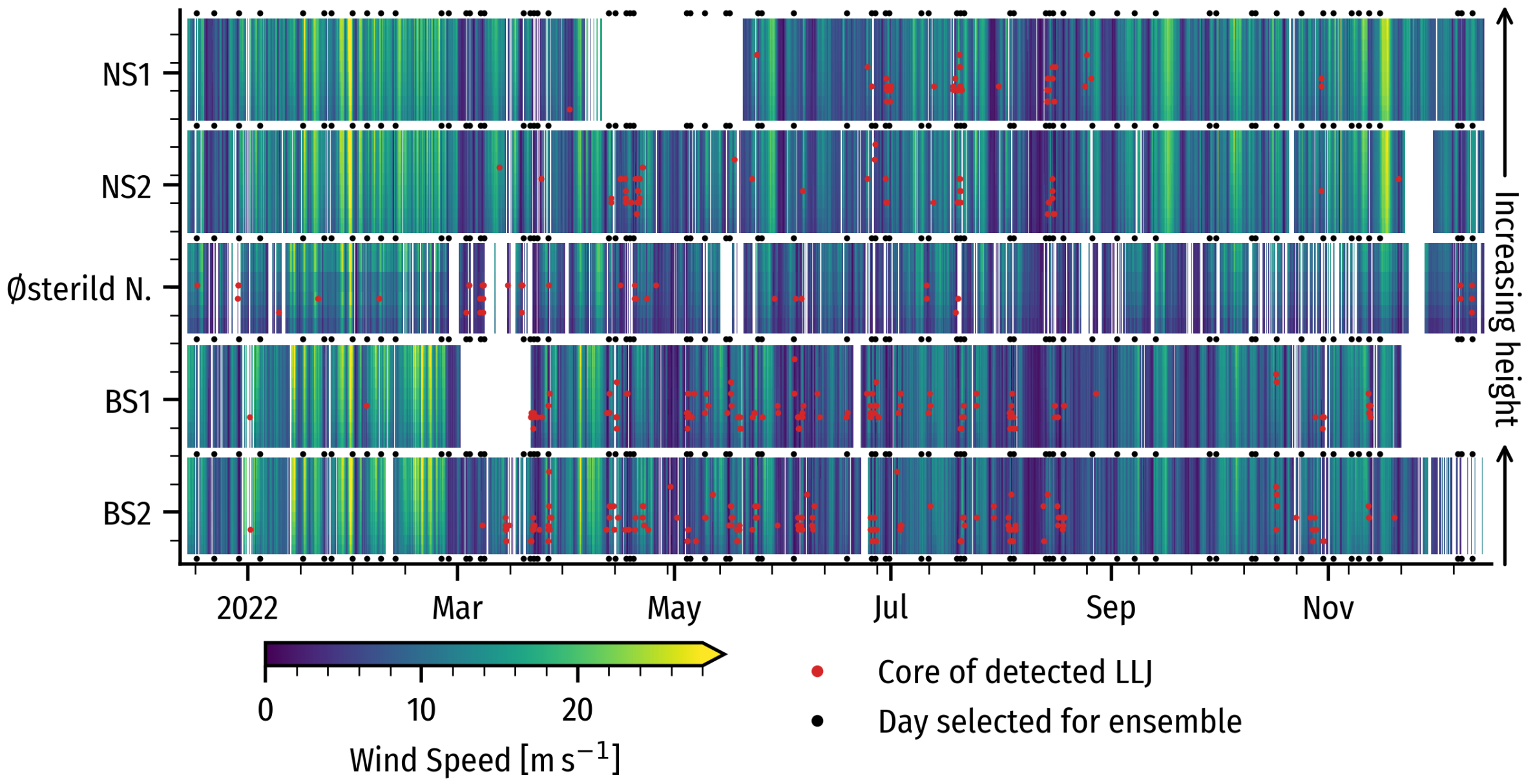

**Figure 4.** Time-height evolution of the wind speed from the measurements at the five sites for the period 15 November 2021 to 15 December 2022. Red dots indicate the core of a detected LLJ. Black dots in the top and bottom margins indicate that the day is part of the WRF model ensemble evaluation. Missing data are left blank.

Østerild N. This means that LLJ heights often fall below the reference turbine's hub height (dashed line), placing the rotor's upper part in the strong negative shear region with strong negative wind shear below that.

Figure 6 shows wind roses for the five sites at 150 m for all conditions and LLJ events. In the North Sea sites, prevalent winds are westerly and north-westerly, but LLJs occur mostly with easterly winds from Denmark and Germany. Østerild N. also sees more LLJs with easterly winds, deviating from the usual westerlies. In the Baltic Sea sites, while westerlies are also the predominant wind direction there, LLJs are more frequently from east and southeast directions, suggesting the influence of a strong air-sea gradient and possibly a coastal baroclinic zone influencing the LLJ formation.

## 4 Model evaluation

We use a 70-day evaluation period and divide the analysis into general conditions, using all the data from the 70-day evaluation period and LLJ-related performance metrics, where we compare modeled vs. observed distribution from all samples with a detected LLJ. This means that the samples are the same for the general metrics and represent different times for the LLJ-related metrics. The hourly model time series were extracted from the grid cell closest to the measurement locations of the same type (nearest offshore point for BS1-2 and NS1-2, nearest land point for Østerild N.).

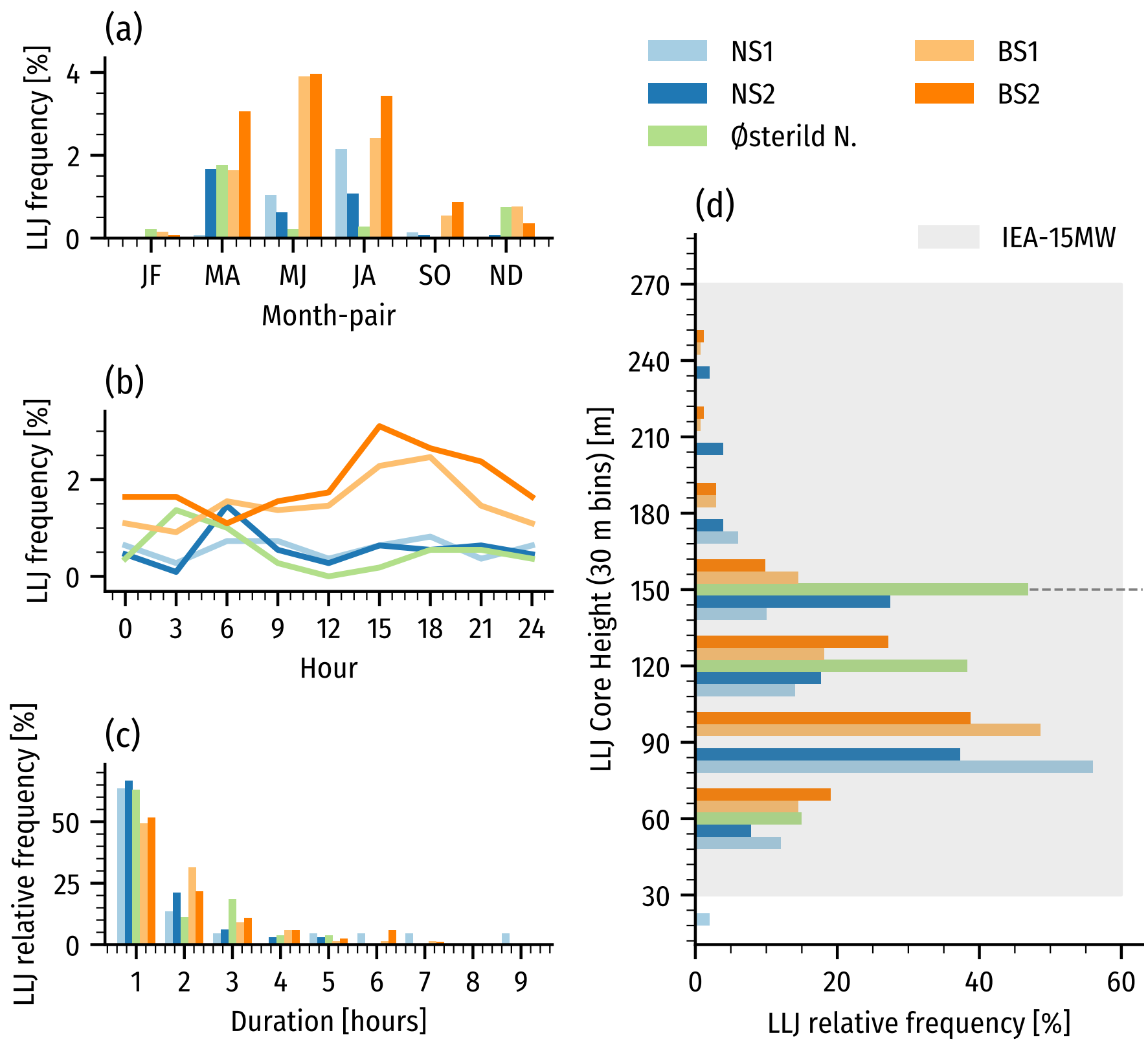

**Figure 5.** Distribution of the occurrence of LLJs as a function of season (a), time of the day (b), LLJ duration (c), and core height from the measurements at the five sites. In panel (d), the span of the IEA 15 MW rotor is shown in gray, with the hub height indicated by a dashed black line.

### 4.1 Selecting the simulation days for ensemble evaluation

The 70 days used to evaluate the model performance are selected from the period 15 December 2021 to 14 December 2022. Of the 70 days, 47 days are selected due to the detection of a strong LLJ (2 m s$^{-1}$ and 20 % falloffs) in at least one of the five sites during that day. To balance the sample of days to include more general weather conditions, 23 additional days are randomly selected, stratified by month, ensuring at least 5 days per month (see Table D1). The chosen days are marked by black dots along the margin of Fig. 4. Although still skewed somewhat towards situations favorable to LLJ development, this more balanced sample of days should improve annual average performance estimates and help assess the simulations' ability to correctly model non-events, i.e., the absence of LLJs when none are observed. Table D1 in appendix D lists the number of days of each month selected from the presence of a LLJ and the number of days selected randomly.

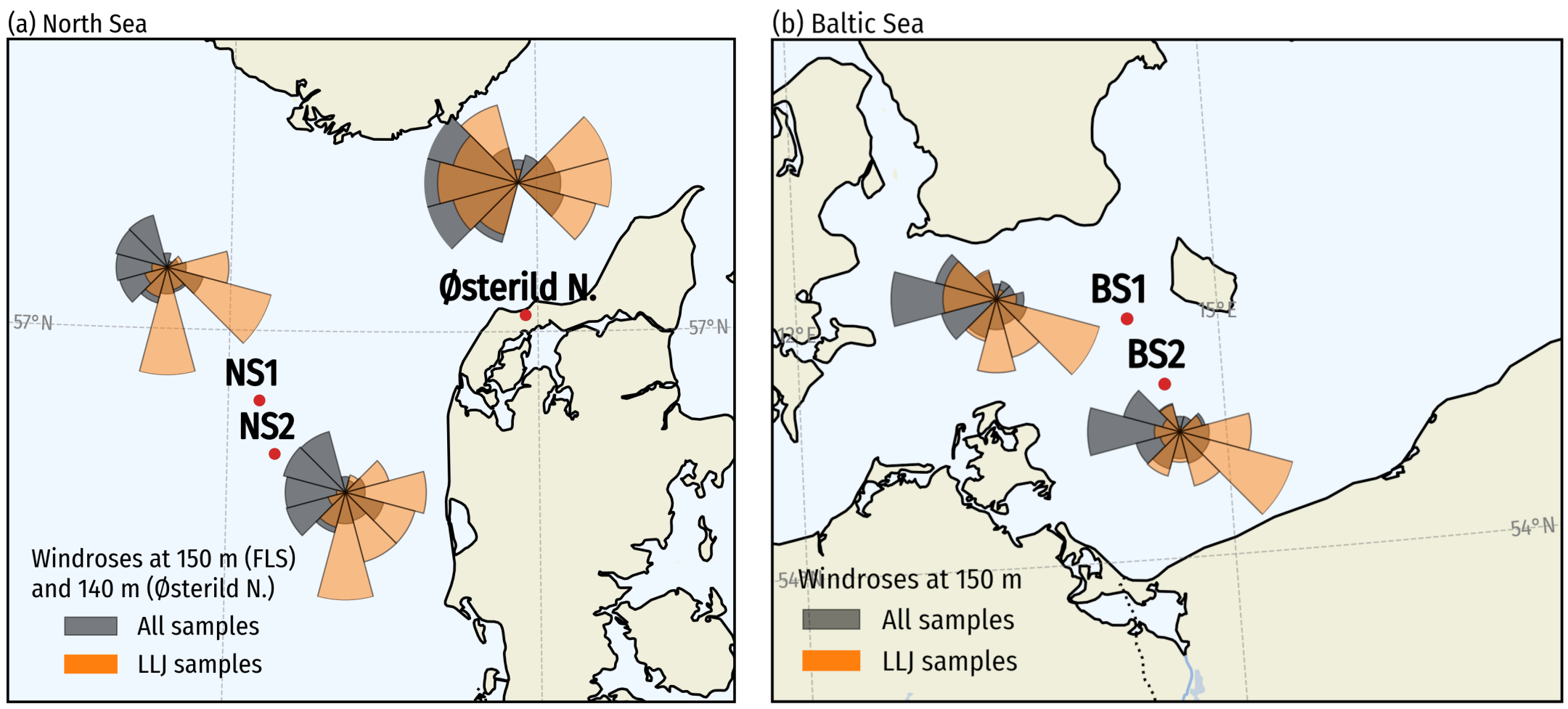

**Figure 6.** Comparison of the wind direction roses for all available samples (gray) and LLJ occurrence periods (orange) at 150 m (FLS) and 140 m (Østerild) at the (a) North Sea and Østerild sites and (b) Baltic Sea sites. Note that to avoid overlap, the wind roses are offset from their geographic location denoted by red dots. The same radial scale is used for the wind roses at each site but may differ between sites.

### 4.2 General model evaluation

Figure 7 displays the mean model performance scores. Compared to ERA5, the WRF model simulations exhibit reduced forecast accuracy, as indicated by higher RMSE(REWS) and lower $R^2$(REWS) values. This is alleviated somewhat by grid-nudging (ensemble member E_MYNN_NUD). On the other hand, several WRF model runs demonstrate enhanced performance relative to ERA5 for scores relating to average quantities and distributions. The largest improvements are seen for the vertical-averaged statistical distances of shear and veer distributions (MEMDS and MEMDV), indicating the vertical structure of the ABL is better captured in the WRF model runs. Notable exceptions to this are ensembles E_BL and E_MYNN_ML0. An improvement in EMD of wind direction is also seen for E_YSU, E_MYJ, and E_3DTKE.

Only one WRF ensemble members, E_MYJ, outperform ERA5 according to the equal-weighted mean rank of scores (Fig. 7i). Conversely, two WRF ensemble members, E_BL and E_MYNN_ML0, perform significantly worse, underestimating REWS by approximately 4 % on average. These members also display poorer distributions of REWS and, as discussed above, MEMDS and MEMDV, resulting in the lowest mean-rank scores. Changing the WRF model setting of the scale-aware mixing length (bl_mynn_mixlength) in the MYNN PBL scheme from option 0 to 2 leads to substantial improvements, with the six versions using E_MYNN performing on par with most other configurations.

Nearly all scores improve, e.g. EMD($WD_{150}$) and MEMDV for the 1 km domains, relative to 3 km domains. The MEMDS and EMD(REWS) also generally improve for most members with a 1 km grid spacing, though some members show slight deterioration. When evaluated using equal-weighted mean rank scoring, the 1 km domains outperform the 3 km domains for eight out of nine WRF ensemble members.

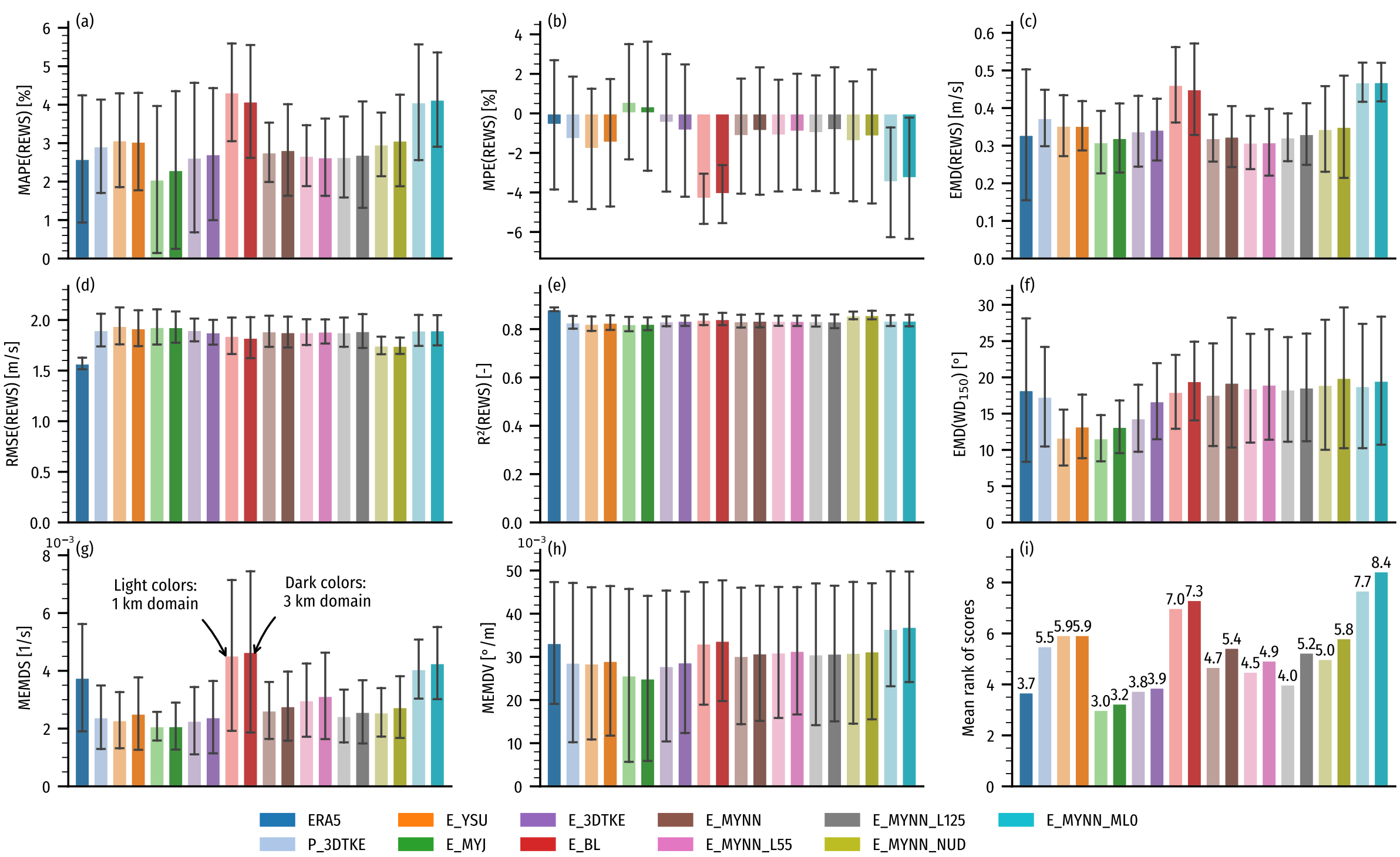

**Figure 7.** Performance scores averaged over the five sites for the 70 evaluation days: (a) MAPE of REWS, (b) MPE of REWS, (c) EMD of REWS, (d) RMSE of REWS, (e) Pearson correlation coefficient of REWS, (f) EMD hub-height wind direction, (g) MEMDS, (h) MEMDV, and (i) the mean rank across all scores. The models are ERA5 (dark blue), the WRF model run chosen for the LLJ climatology (P_3DTKE; light blue), and the different WRF model ensemble members from Table 2. The error bar represents the spread among the five sites. For each WRF model ensemble, the light color is for the 1 km domain (D3-E – D5-E); the darker color the 3 km (D2-E) domain as shown in panel (g).

The effect of varying the number of vertical model levels on the evaluation statistics is not very pronounced, though differences do arise. Among the ensemble members, E_MYNN_L125 with 125 vertical levels and a 1 km grid achieves the best overall performance of the MYNN-based members. Despite this, individual scores for the evaluated metrics remain similar across different vertical-level configurations.

The MEMDS and MEMDV metrics hide a lot of details. The primary source of MEMDS error for all the models is an underestimation of the wind speed shear at lower levels (below 100 m) at all sites, in particular, ERA5, E_BL, E_MYNN_ML0, and E_MYNN_L55 have a strong underestimation, while E_MYJ is most accurate. Higher up (above the turbine hub height), the shear is generally weaker and captured better by the ensemble members, even though most continue to underestimate the shear there, E_YSU, E_BL, and especially ERA5, overestimate it. For wind veer, all models show an underestimation at lower levels, except for E_MYJ, which overestimates it.

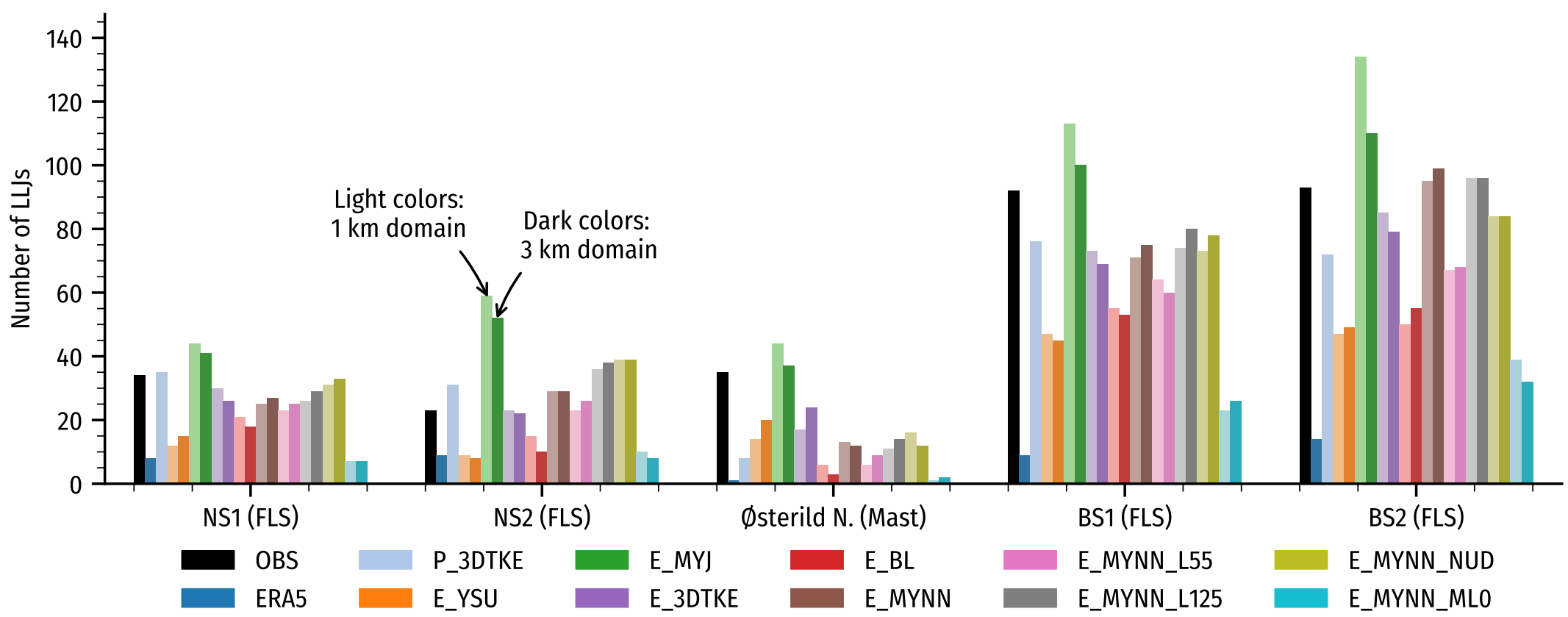

**Figure 8.** Number of detected LLJs at each site during the evaluation period.

Considering only the standard statistics in Fig. 7, the E_MYJ ensemble will be considered the best-performing WRF model ensemble member. The mean rank of the scores is lower than any other member.

### 4.3 LLJ-related model performance

Figure 8 shows the total number of LLJs detected at each site for each model and in the observations during the 70-day evaluation period. In Fig. 9, the results for the ten different LLJ-related performance scores, focusing on average errors and statistical distances between distributions, are presented. They highlight significant differences between the WRF model runs in their ability to correctly simulate LLJs and model their characteristics. For the Østerild N. site, several of the models detected fewer than 7 LLJs (including ERA5). Thus, for all the average scores in Fig. 9, except for MAPE and MPE of LLJ rates, only the four other sites were used in the calculation for all the models.

Some of the most notable results for LLJ-related model performance are:

- The E_MYJ ensemble member significantly overestimates the mean LLJ rate (especially in winter; not shown), while ERA5, E_YSU, E_BL, and MYNN2_ML0 substantially underestimate it.
- The ERA5 not only underestimates LLJ rates but also overestimates the LLJ core height resulting in an average bias of ≈45 % (not shown) and the poor EMD score shown in Fig. 9c. The ERA5 also fails to capture the annual and diurnal cycles of LLJs accurately. ERA5 also performs significantly worse for distributions of jet wind speeds and the shear and veer distributions, MEMDS, and MEMDV.
- The E_YSU ensemble member captures best the distribution of LLJ core heights (Fig. 9c). The EMD errors shown in the figure for jet core heights come from a general overestimation of the height of the LLJ peak by ≈10 % for most WRF model members, E_YSU at ≈5 %, and E_BL at (≈20 %).

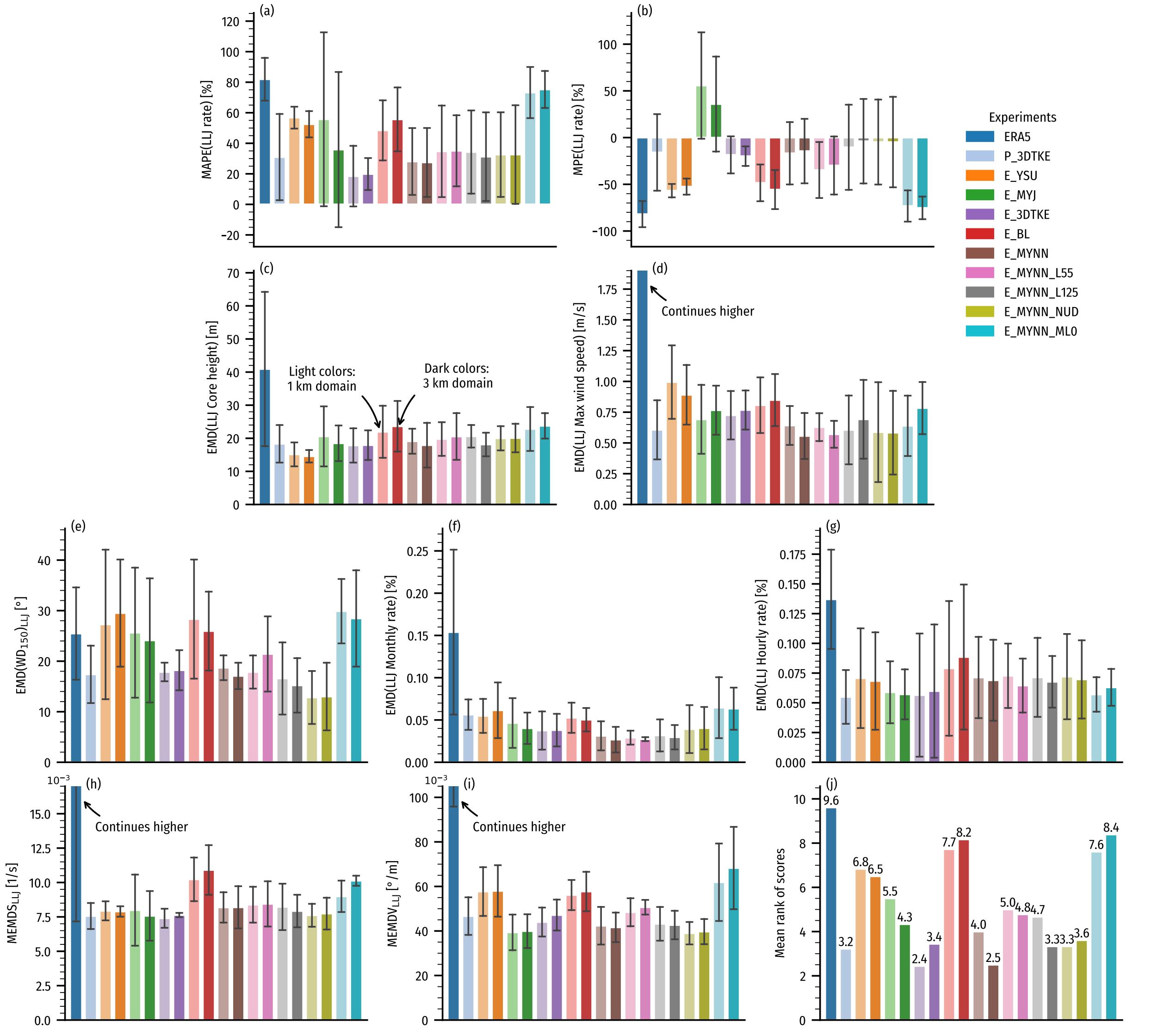

**Figure 9.** Performance scores aggregated over the five sites for the observed vs. modeled rates and distributions of LLJ-related quantities during LLJ events detected during the 70-day evaluation period: MAPE of LLJ rate (a), MPE of LLJ rate (b), EMD of LLJ Core height (c), LLJ core wind speeds (d), EMD of hub-height wind roses (e), EMD of monthly LLJ rates (f), EMD of hourly LLJ rates (g), MEMDS (h), and MEMDV (i), and the mean rank across all scores (j). The bar represents the spread among the five sites. For each WRF model ensemble, the light color is for the 1 km domain (E3-E – D5-E); the darker color is the 3 km (D2-E) domain as shown in panel (g). Because too few LLJs were detected for some of the models at Østerild N., the scores in (c)-(j) are based only on NS1, NS2, BS1, and BS2 for all models.

295 – The WRF ensemble generally overestimates the LLJ wind speeds at the LLJ core by ≈5 % on average, less so at BS1 and BS2 (0 % to 5 %). The E_MYNN_ML0 ensemble member is an outlier from the rest. It underestimates the wind

speed (≈−3 % on average. The best performing ensembles, in terms of EMD(LLJ Max wind speed), are the E_MYNN members (both the 85 and 125 levels version)

- The E_3DTKE captures well the LLJ occurrence rates diurnal and annual cycles. Its annual cycle of LLJ occurrence is consistently accurate at all five sites. The other WRF ensemble members also capture the diurnal cycles well but are less accurate at other sites, mostly NS2. E_MYNN_ML0, again, stands out negatively with the lowest scores for the diurnal cycle. With the exception of the E_MYJ, which consistently overestimates the LLJ rate in all seasons, all WRF ensemble members underestimate the fall and winter rates and overestimate the summertime rates (not shown). Most models capture the diurnal cycle well at the BS1 and BS2 sites (E_3DTKE has the lowest errors there), but score lower at especially BS2.

- The distributions of shear and veer during LLJ events averaged across rotor-heights (as measured by MEMDS by MEMDV) are captured about equally well by several WRF members (E_YSU, E_MYJ, E_3DTKE, E_MYNN, and E_MYNN_NUD). E_3DTKE is slightly better for shear and E_MYJ for veer. E_BL and E_MYNN_ML0 stand out with the worst scores. This comes from large underestimations of the shear across several levels, compared to the other ensemble members.

  The performances of the 3 km WRF domains are generally not significantly worse than the 1 km domain for LLJ-related metrics, with the rank-average scores only improving for four out of nine ensemble members.

Among the ensemble members, the E_3DTKE emerges as the best performer, accurately capturing the mean LLJ rate, seasonal and diurnal trends, and vertical shear and veer structure. Although the E_MYJ also captures the seasonality and diurnal structure well, it tends to exaggerate the occurrence of LLJs and has a less accurate wind rose compared to E_3DTKE. Consistent with their performance in the general metrics, E_BL and E_MYNN_ML0 produce the worst LLJ-related scores overall. In contrast, E_MYNN, the baseline setup using MYNN, generally scores high on most of the metrics. Interestingly, there are no significant differences between the three ensembles with different numbers of vertical levels in the E_MYNN configuration for many of the metrics. A slight deterioration vs the baseline MYNN is seen for both L55 and L125 when looking across all metrics, indicating that the number of vertical levels investigated here has a minimal impact on LLJ-related performance metrics.

### 4.4 Spatial LLJ variation model evaluation

Figure 10 illustrates the mean LLJ occurrence rate across the various ensemble members in Table 2 during the evaluation period. For these maps, all model levels up to 500 m are used. All maps exhibit consistent spatial patterns, with higher occurrence in the Baltic Sea around Bornholm and Northern Germany near Hamburg. However, absolute LLJ jet frequency varies significantly among the models.

The E_MYJ ensemble member stands out, showing the highest rates, with values exceeding 28 % across substantial regions of the Baltic Sea near Bornholm and along the Swedish east coast. Conversely, the lowest rates are produced by ERA5, NEWA,

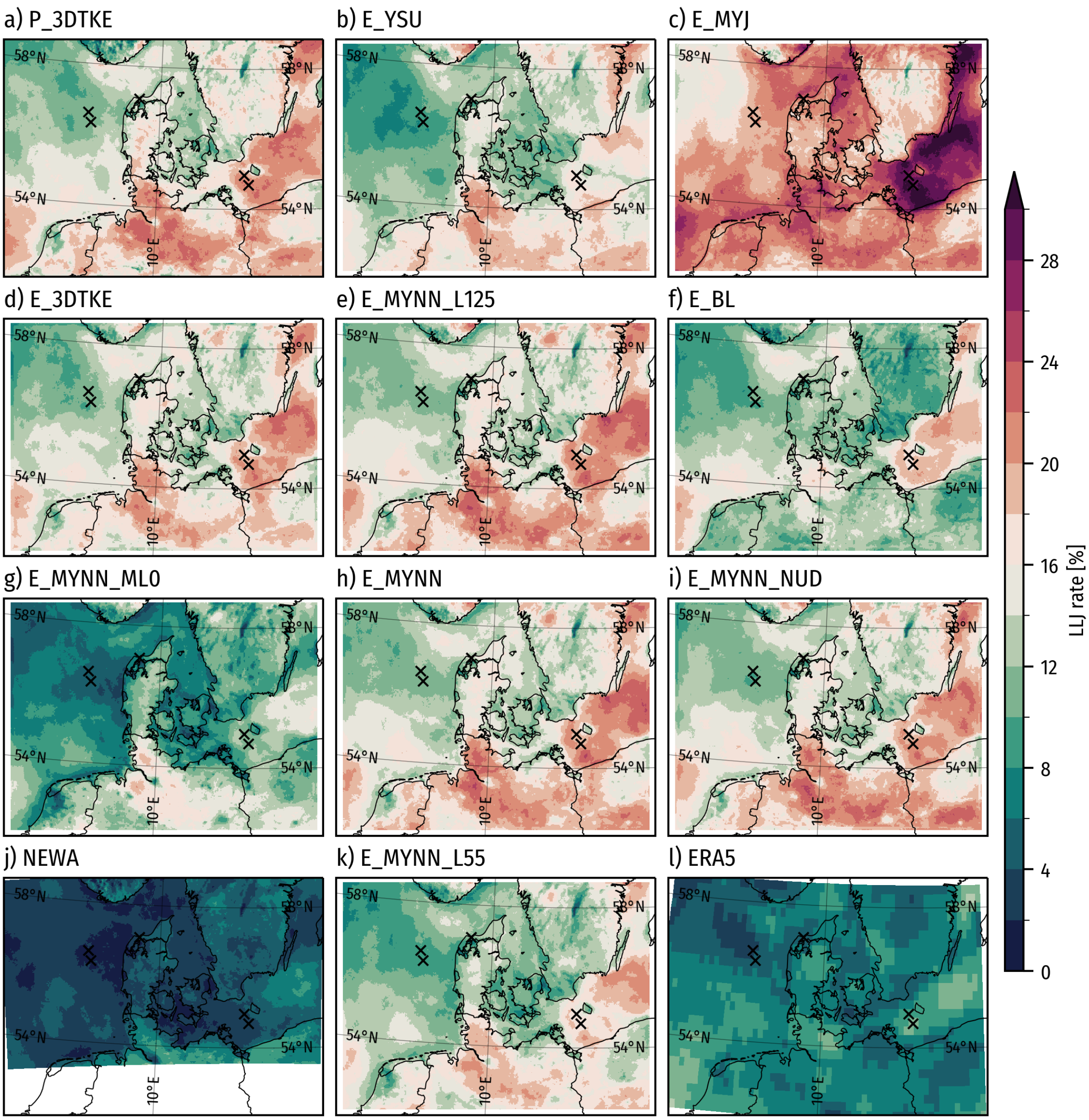

**Figure 10.** Maps of LLJ occurrence rates [%] during the 70 evaluation days: (a)–(i) and (k) WRF ensemble members in Table 2, (j) NEWA, and (l) ERA5. The black crosses show the measurement sites' location.

and E_MYNN_ML0. The other ensemble members present rates that are relatively similar to each other. In the case of the E_MYNN member, an increase in the number of vertical levels in the simulation notably enhances the LLJ rates, particularly when the levels are raised from 55 to 85 (Figs. 10 (e), (h) and (k)), but much less from 85 to 125.

While the absolute LLJ occurrence rate is important, it is quite sensitive to the detection window and thresholds chosen for their detection. To evaluate the relative spatial variations of LLJ rates we compute the spatial Z-scores as shown in Fig. 11. The maps reveal striking similarities of spatial variation, but also member-specific variations. While all the models show LLJ "hot spots" in the Baltic Sea around Bornholm and along the Swedish east coast as well as around Hamburg in Northern Germany, some models make the Baltic Sea rate more pronounced (in these relative Z-score terms), including E_BL and E_MYJ. The same members also have stronger LLJ rates in the near-shore areas in the North Sea, e.g. Kattegat, Wadden Sea, and the Southern Bight. The influence of the Island Bornholm varies as well, with some models showing a stronger island-to-sea gradient in LLJ occurrence rates, including E_YSU, E_3DTKE, and the MYNN-based members, in contrast to E_MYJ and E_BL. In the North Sea, the models agree that a tongue of low rates extends from the West coast of Jutland into the central North Sea, but with member-specific variations: in E_YSU, E_3DTKE, and the MYNN-based scenes it is more pronounced than E_MYJ and E_BL. Onshore, Northern Germany is more pronounced in E_MYNN_ML0. The NEWA and ERA5 grids are different from the WRF ensemble domains and they have different vertical levels, this influences the results somewhat, but strong similarities in the spatial variation are still visible. In E_BL and ERA5, the North Sea close to the Strait of Dover is more pronounced than in the maps from the other models. Although the absolute levels are different, the spatial distribution of LLJ rates of ERA5 is consistent with those presented in e.g. (Rubio et al., 2022) (strong LLJ prevalent in the sea south of Götland).

### 4.5 Evaluation of the climatological run

The WRF model simulation used for the LLJ climatology, P_3DTKE, uses the same physics parameterization as E_3DTKE, but for computational efficiency, it uses longer simulation times (7 days vs 1-day simulation used in the evaluation part), and the larger domain (Fig. 1), and grid-nudging above level 50, which lies around 2000 m above ground level. These changes were necessary to include most of the North and Baltic Sea regions and to increase computational efficiency (reduced number of spin-up hours per simulated day). We used two simulations to evaluate the effect of the length of the simulations, S_3DTKE_3D and S_3DTKE_7D, with the same configuration of E_3DTKE, but different lengths and spin-up (see Table 2). The metrics for S_3DTKE_3D are slightly higher in general ranking to those for S_3DTKE_7D; the LLJ ranking is identical, but slightly worse than that of E_3DTKE.

While there are some changes in evaluation metrics for P_3DTKE relative to E_3DTKE, such as reduced overall scores in the general model evaluation (Fig. 7), the changes are not drastic, and the runs retain good performance metrics for LLJ-related scores (Fig. 9). The spatial distribution in LLJ occurrence rates also remains fairly consistent with E_3DTKE (Fig. 10 and 11). Incremental tests of the configuration showed that the biggest reduction in performance scores came from adding nudging and using the large domain and less from integration time.

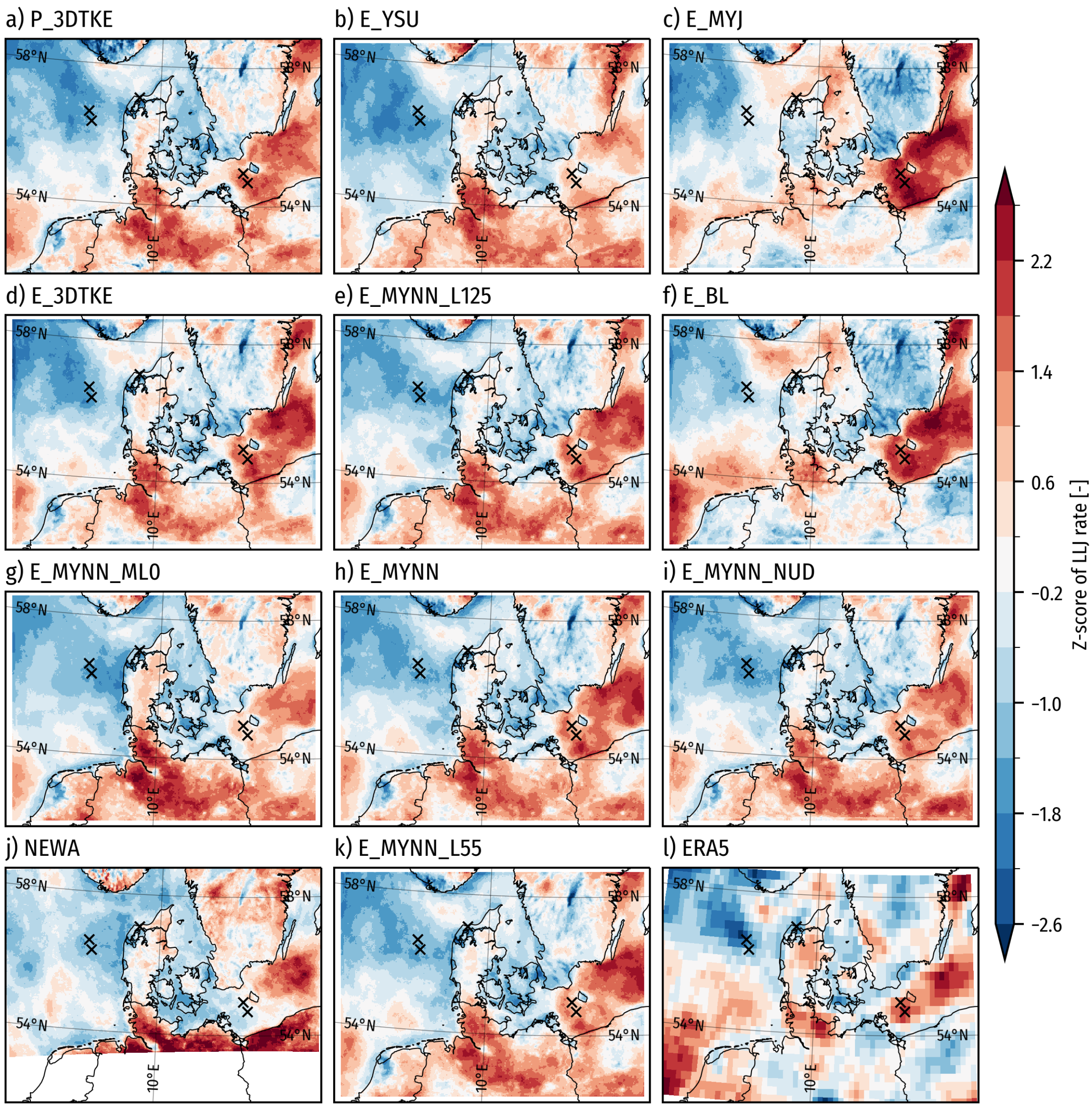

**Figure 11.** Maps of normalized LLJ rates (Z-scores) during the evaluation period for: (a)–(i) and (k) WRF ensemble members in Table 2, (j) NEWA, and (l) ERA5. The extent of maps for NEWA and ERA5 are slightly different compared to the WRF ensemble. The black crosses show the measurement sites' location.

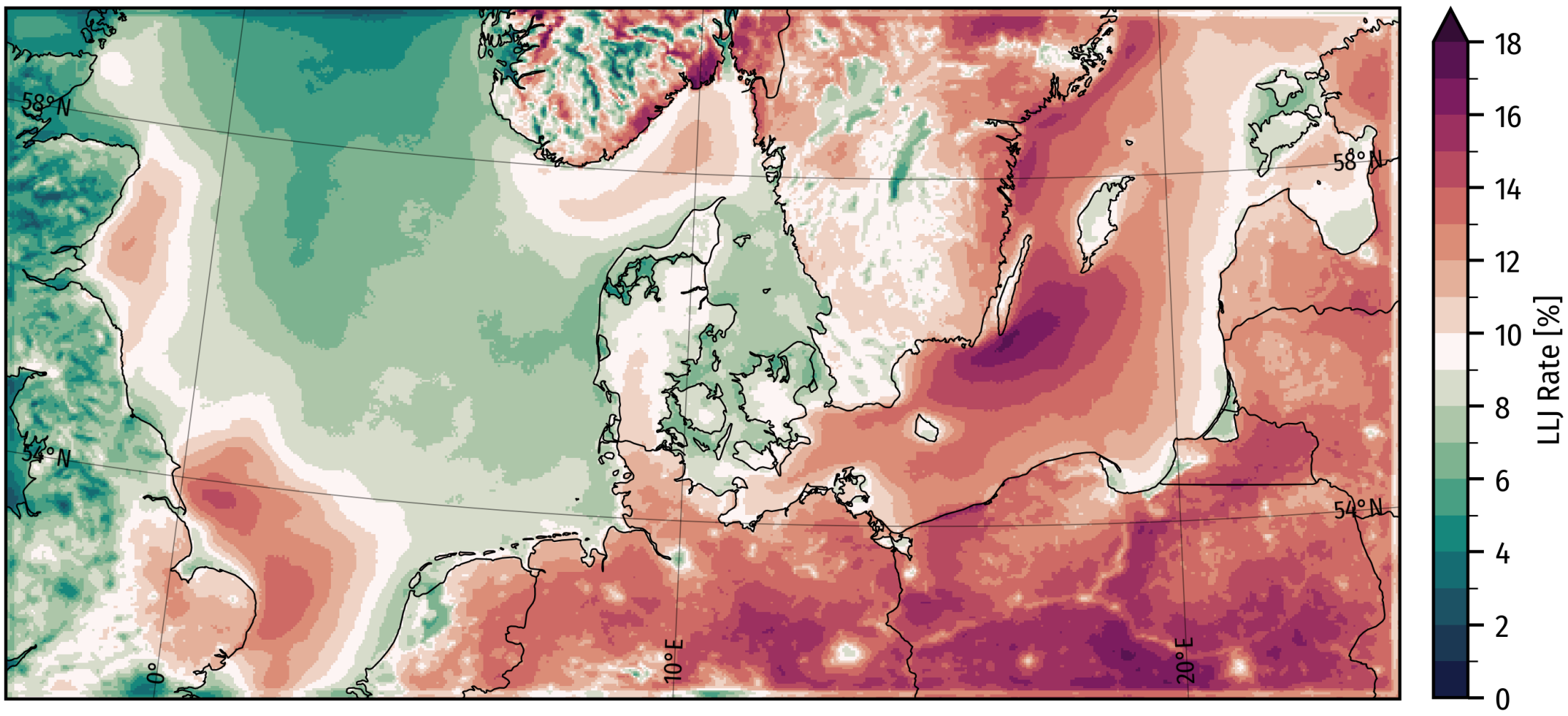

**Figure 12.** LLJ occurrence rates [%] in the WRF model climatology from 26 June 2019 to 26 June 2024.

## 5 LLJ Climatology

The P_3DTKE WRF model simulation spans the five years from 26 June 2019 to 26 June 2024. To create the climatological layers presented here, we use all output from the WRF model domain D2-P (after removing the spin-up period) at 30 min intervals, in the same manner as was done in the NEWA simulations (Dörenkämper et al., 2020). Here we present various aspects of the LLJ characteristics. The strong seasonality of the LLJs means that most quantities are presented as seasonal aggregates.

### 5.1 Occurrence rates

Figure 12 shows the spatial distribution of annual mean LLJ occurrence rates for the five-year climatology in the larger domain. It highlights significant LLJ prevalence south of Öland in the Baltic Sea, onshore in central Europe, and along eastern-facing coastlines, particularly in Sweden and the UK. Skagerrak shows higher LLJ prevalence than Kattegat and the Danish North Sea. Smaller islands, urban areas, lakes, and rivers generally have lower LLJ rates than the surrounding areas, evident in the large Swedish lakes Vänern and Vättern, the islands of Bornholm and Götland, the Warta, Noteć, and Vistula rivers in Poland, and urban centers around Berlin, Hamburg, and Warsaw, which all clearly stand out. Mountainous regions exhibit higher local spatial variation in LLJ rates. The parts of the North Sea furthest from shore, including Dogger Bank, show lower LLJ rates than most coastal areas.

The contrast in occurrence rates for the longer P_3DTKE simulation against the rates of the 70-day periods for E_3DTKE (Figure 10), points to some remaining skewness towards LLJ weather conditions in the 70-day sample. The higher rates of LLJ occurrence in the Baltic Sea are quite similar in the long and short simulations; however, the concentration in LLJ rates

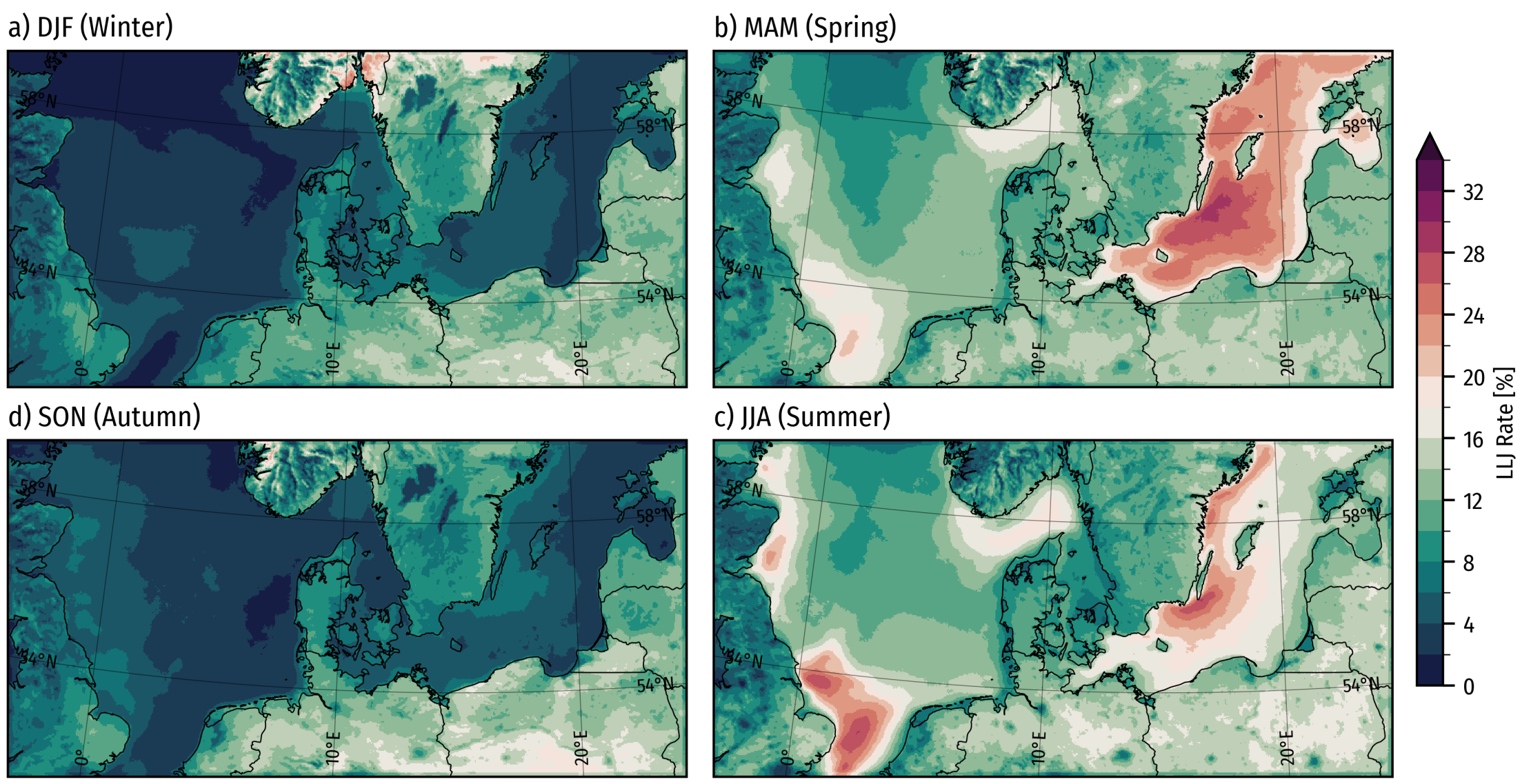

**Figure 13.** LLJ occurrence rates in the WRF model climatology (June 2019 to June 2024) separated by season.

around Hamburg (especially North of the city) seems less pronounced in the long WRF model simulation than in some of the ensemble ones. Instead, we see a more broad onshore area with higher rates in the Northern parts of Central Europe.

Figure 13 shows that offshore LLJ occurrence rates exhibit strong seasonality, while rates over land remain more constant throughout the year. In spring, the Baltic Sea has a high prevalence (over 20 %), especially southeast of Öland, extending from Denmark to the Bay of Finland. In summer, LLJ prevalence remains high in these areas, particularly south of Öland, with a slight reduction elsewhere. The seas east of the UK show high LLJ rates in spring and summer, notably in the Southern Bight and around Norfolk Banks and Silver Pit. Skagerrak also sees higher spring and summer LLJ concentrations than other parts of the North Sea and Kattegat. In fall and winter, offshore LLJ rates drop significantly (6 % or less). Onshore, LLJ rates stay close to the annual average (10 % to 20 %), slightly higher in summer and fall over northern Central Europe.

### 5.2 Annual and diurnal patterns

Figure 14 shows the most prevalent month of the year for LLJ occurrence. When the rate for the month in a particular area is 50 % above the equal rate (1/12) the area is hatched with black lines. When the rate is above 100 % more, it is hatched with black dots. The figure reflects how seasonal offshore LLJs are, with rates peaking in May and June. Onshore, LLJ occurrences are not strongly seasonal. However, even though the onshore LLJ rate is fairly consistent throughout the seasons, one must expect that a seasonally varying mix of driving mechanisms is responsible for LLJ formation.

Figure 15 shows maps of the most prevalent hour of the day for LLJ occurrence in each season. Again, with stripe hatches for areas 50 % above the equal rate (1/24), and dots for 100 % above. Onshore, LLJ occurrence is highly tied to the time of the day, with peaks happening during nighttime. In summer the peak is around midnight, while the peak happens later in the early

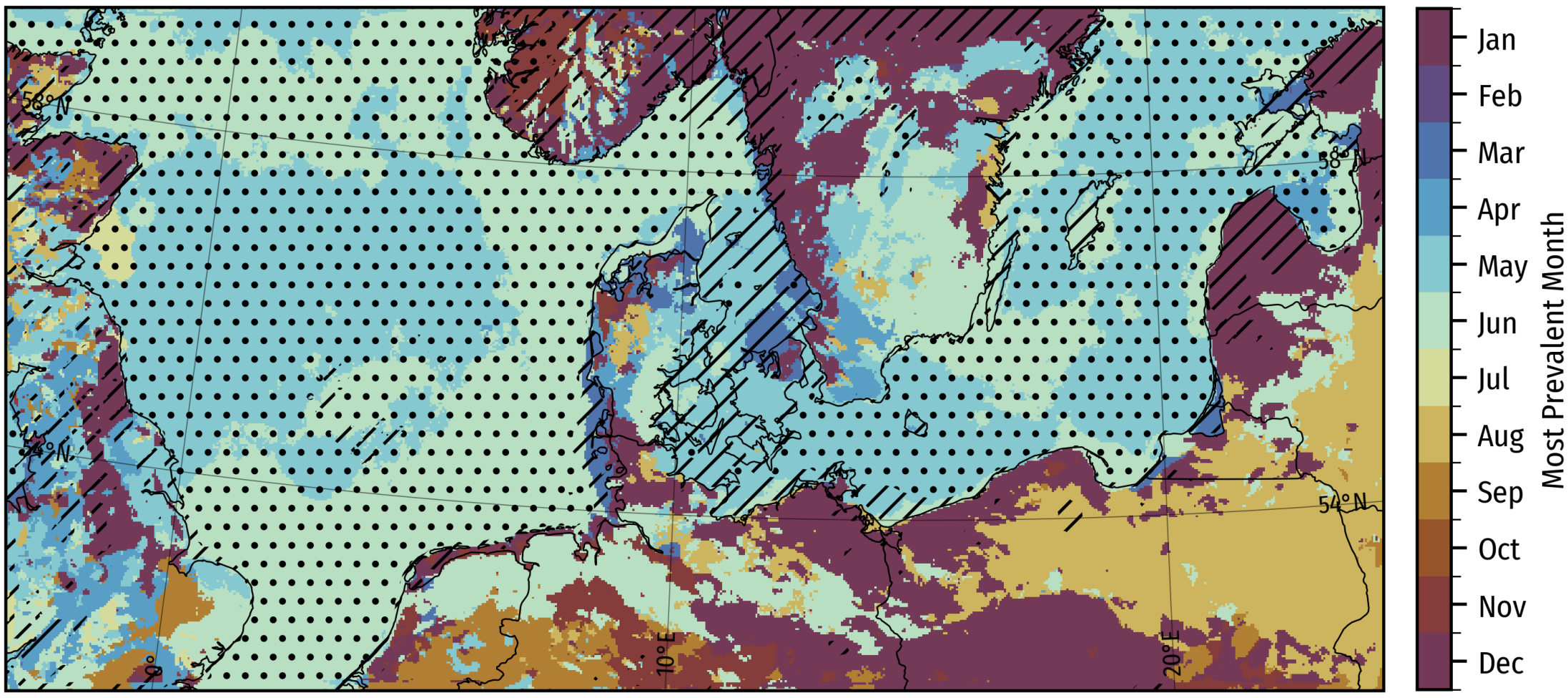

**Figure 14.** The most prevalent month of the year for LLJ occurrence in the WRF model climatology (June 2019 to June 2024). Black lines and dots indicate that the month is respectively 50 % and 100 % more prevalent than the equal rate (1/12).

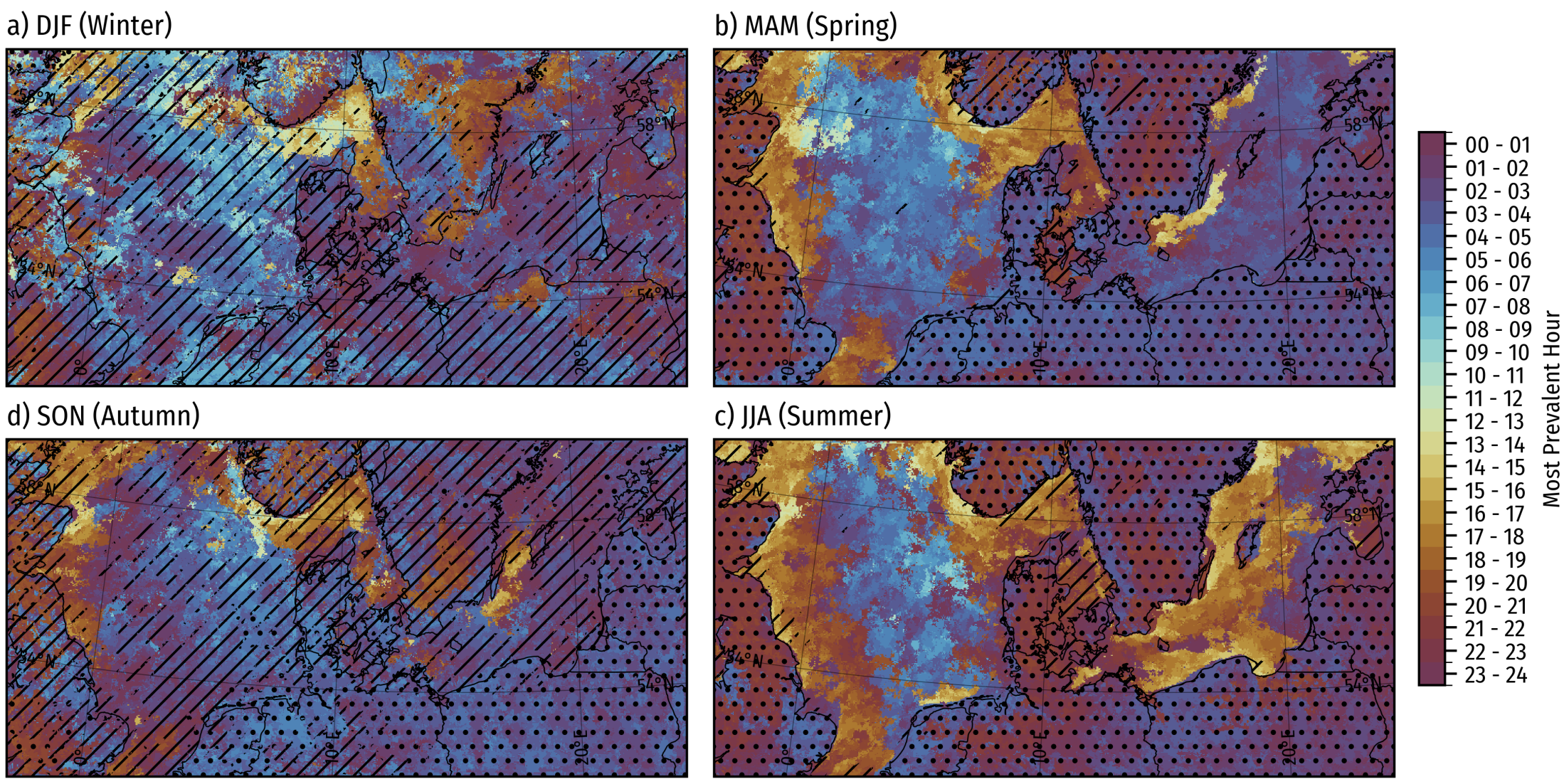

**Figure 15.** The most prevalent hour of the day for LLJ occurrence in the WRF model climatology (June 2019 to June 2024) separated by season. Black lines and dots indicate that the hour is respectively 50 % and 100 % more prevalent than the equal rate (1/24). Hours are in Coordinated Universal Time (UTC).

morning hours in other seasons. Offshore, in the spring and summer seasons, the most prevalent hours are not as significantly defined as onshore. However, some clear patterns still emerge. Along coastlines, the peaks in LLJ rates happen in the afternoon and early evening hours, with some clear temporal evolution with distance to shore, so further offshore corresponds with later LLJ peak hours (see e.g. in the Baltic Sea and along the UK east coast). Another pattern in the seasonal maps is that the peak

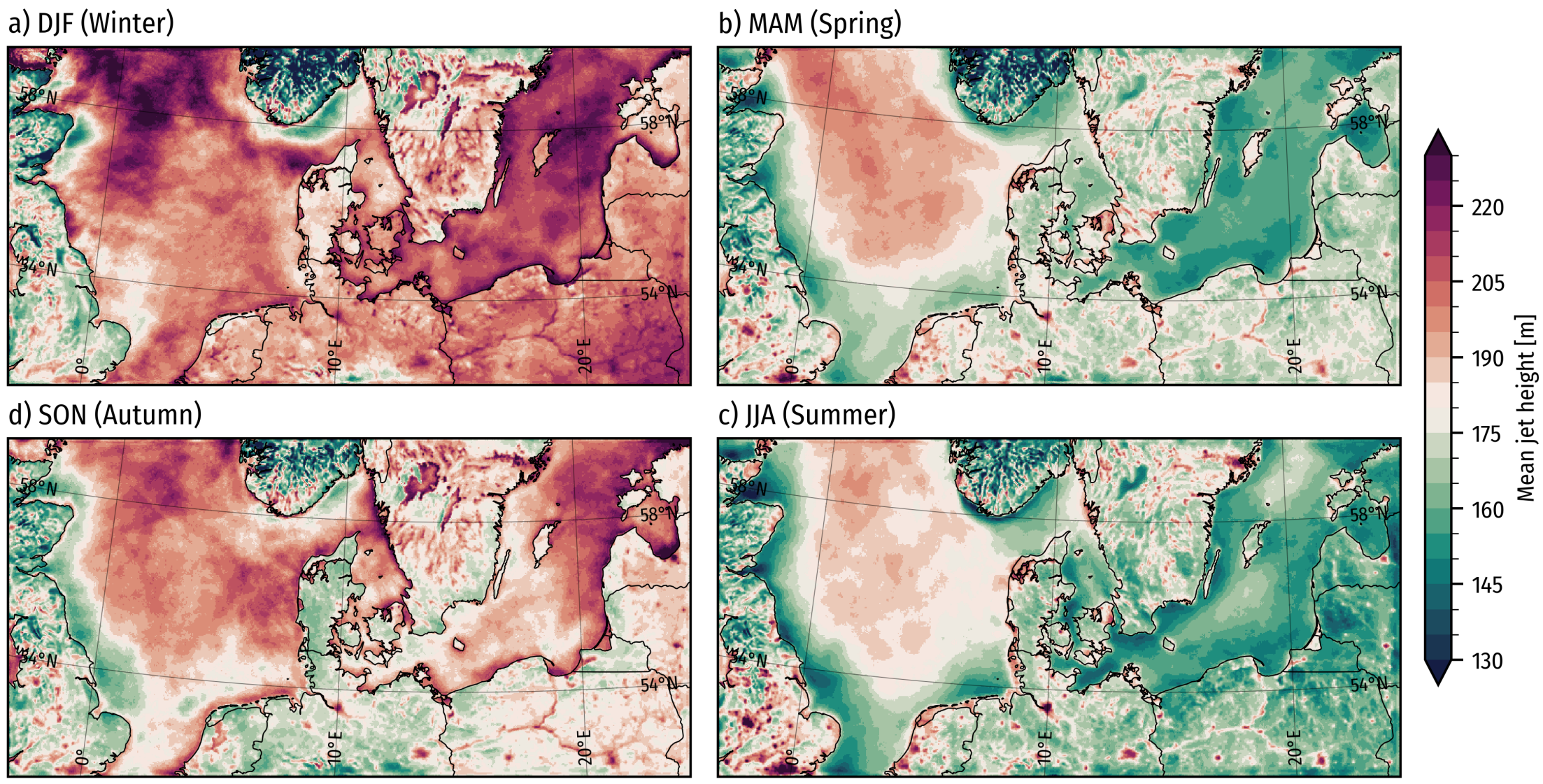

**Figure 16.** Mean height of LLJs in the WRF model climatology (June 2019 to June 2024) separated by season.

occurrence in the middle of the North Sea occurs in the morning hours, and not in the afternoon and evening as could be expected.

### 5.3 Mean core heights

Figure 16 shows maps of the mean LLJ core height for the four seasons. The heights are highest offshore in the fall and winter and furthest from the coast. The lowest mean heights happen in the spring and summer, especially along the coasts where occurrence rates are also higher. In general, an inverse relation exists between the LLJ rates and the LLJ heights. The seasonality in heights is much less pronounced onshore but is lowest in the summer across most of north-central Europe, the Baltic states, and Scandinavia. The underlying distributions of LLJ heights are slightly positively skewed, meaning the bulk of events take place below the mean but with a longer tail of higher LLJs. This underscores that many of the LLJs happen at or below the typical hub height of large offshore turbines, especially along the coasts in the summertime.

### 5.4 Mean duration

The mean duration of LLJs is presented in Fig. 17 and shows that LLJs are often short-lived, typically lasting no more than a couple of hours. However, the offshore spring and summertime LLJs occurring in the Baltic Sea and along the east coast of the UK tend to last longer on average. The longest-lasting LLJs are seen in Southern Bight and the English Channel, where a few very long-lasting jets increase the mean duration. The distribution of LLJ duration is generally “short-heavy” in most places but with long tails of long-duration events of lower probabilities. In central Europe, a band of higher-duration LLJs appear in

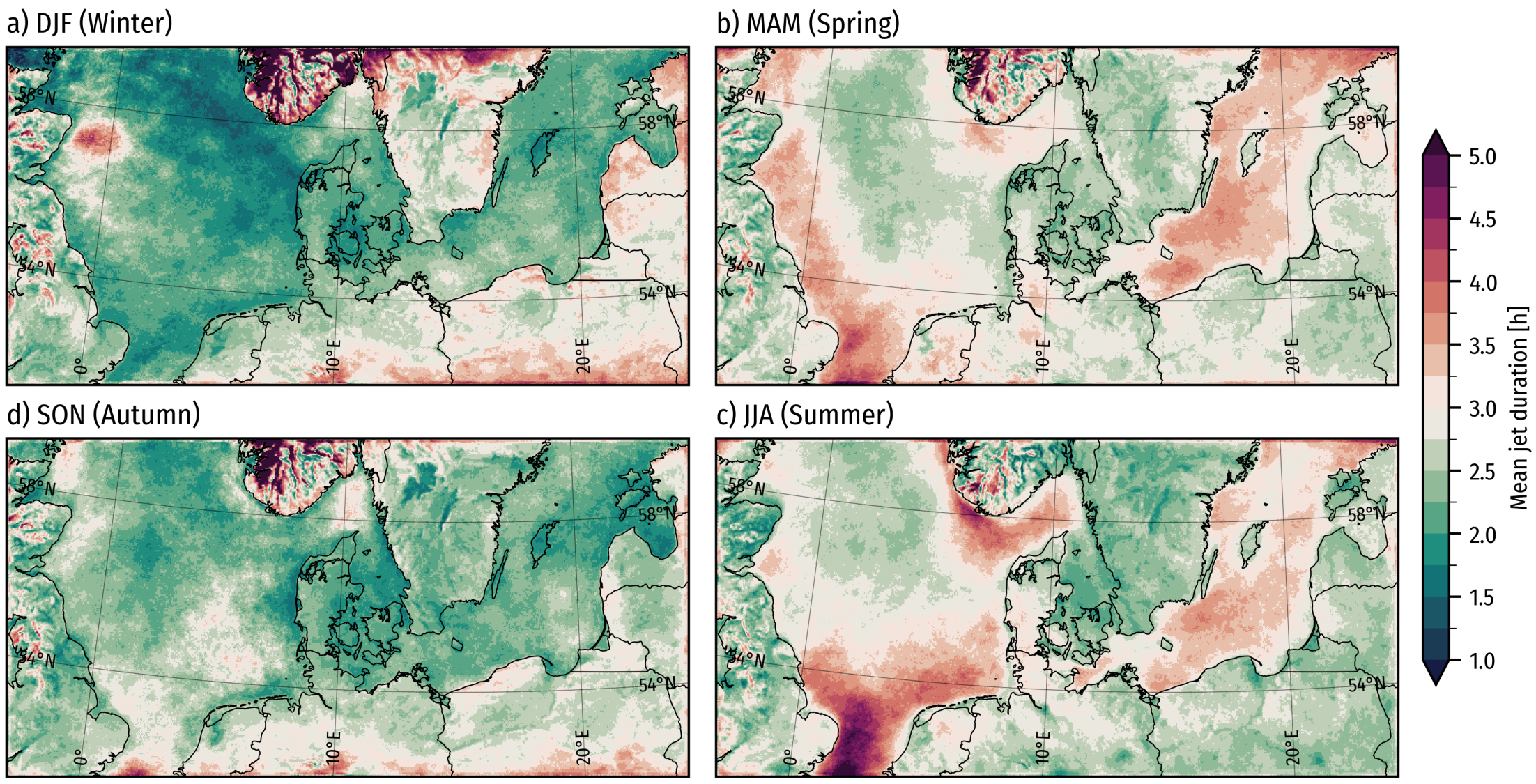

**Figure 17.** Mean duration of LLJs in the WRF model climatology (June 2019 to June 2024) separated by season.

the fall and winter. Interestingly, an area of longer mean duration is present in winter offshore east of the Scottish mainland, possibly indicating some form of interaction between LLJ events and the inland terrain.

## 5.5 Jet magnitude and direction

In Figure 18 we present the seasonal maps of the average relative wind speed falloffs from detected LLJ cores to its minimum above (colors), with the circular-average wind directions of the detected LLJ cores indicated with black arrows. Because the value of the surface roughness length strongly influences the falloff and shear below the maximum, we use the fall-off above the maximum as a proxy for “LLJ strength”. During Fall and Winter, the strong jets, in relative terms, appear close to geographic features, such as the Norwegian mountains (in Skagerrak), hills in Southern Germany, and the Grampian mountains in Scotland. The jet direction tends to follow the predominant wind direction in the fall and winter (mostly southwesterly flow). During spring and summer, the offshore coastal regions have the strongest relative falloffs from the jet core to the minimum above, indicating the most intense LLJs. The directions of LLJ cores in these near-coastal offshore regions point parallel to the coast with the landmass to the left and the ocean to the right, which would be consistent with coastal LLJs formed in a low-level baroclinic zone driven by a land-sea temperature contrast. The prevalent directions match well the wind roses observed during LLJ events at the four FLS sites (Fig. 6), on average winds come from the southeast south of Bornholm for LLJ situations and tend to come from east and southeast at the North Sea sites during Spring and Summer. The weakest falloffs offshore are furthest from shore, for example in the middle of the North Sea.

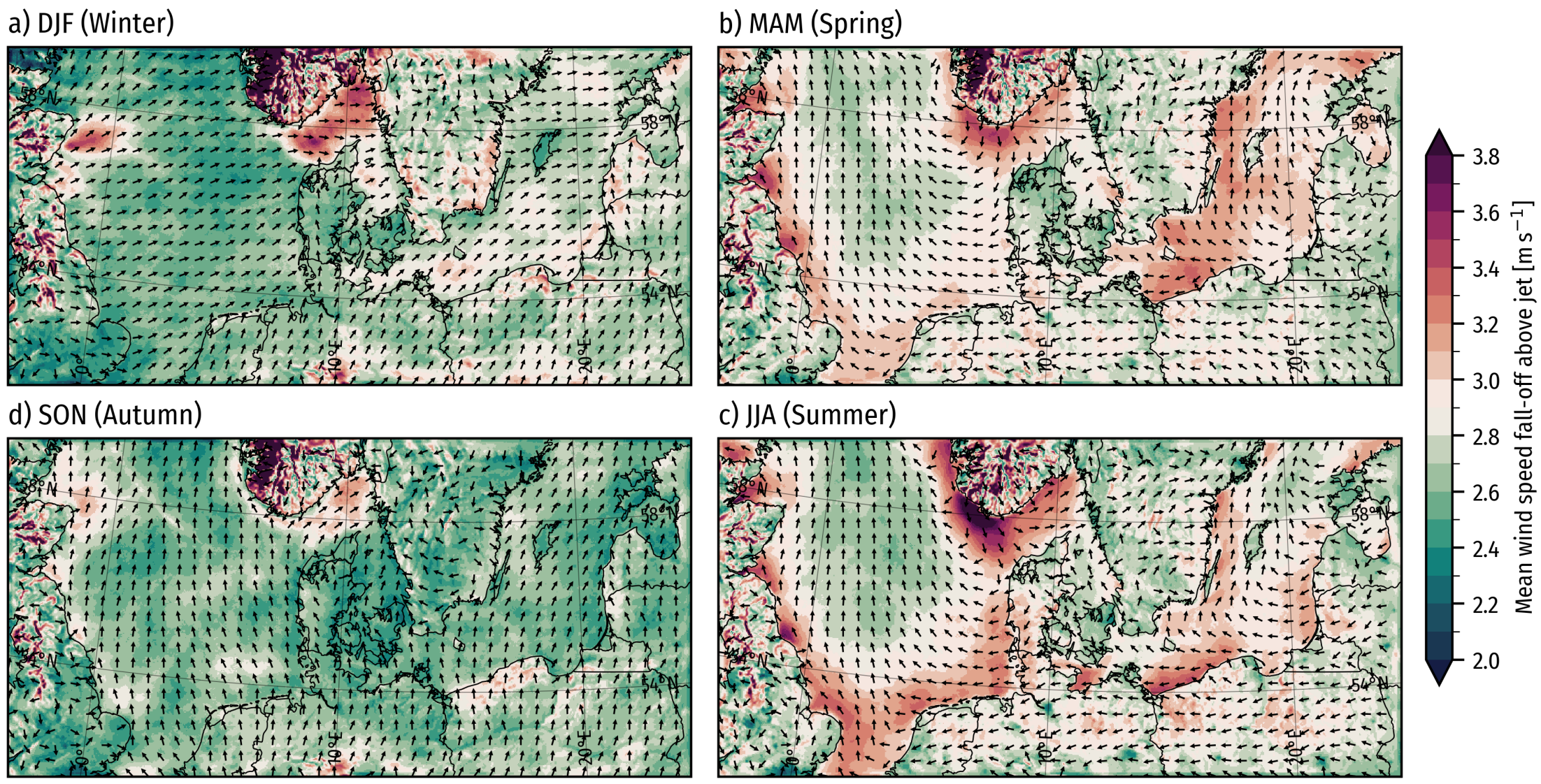

**Figure 18.** Mean wind speed fall-off above the LLJ peak in the climatology (June 2019 to June 2024) separated by season. Black arrows indicate the (circular) mean direction of the jet cores.

### 5.6 Wind energy resources

435 The power capacity factor for the IEA-15MW is significantly higher in the windy fall and winter seasons (Fig. 19), while less energy is available in spring and summer. This makes LLJ events a large share of the total capacity in the latter two seasons, evident in Fig. 20, which shows the share of the total capacity happening during LLJ events in each season. This figure can be compared with Fig. 13, the occurrence rate of LLJs, to see whether LLJs serve to lift the capacity factor or not. From this, we see that LLJs are most important in increasing the capacity factor during the summer and spring (when the overall capacities are lowest). They are also an important source of energy during summer in central Europe. In mainland Denmark, LLJ does not play a big role in the total capacity, suggesting that LLJ events tend to happen during weather patterns associated with lower-than-usual wind speeds in this area.

## 6 Discussion

Our model evaluation shows large differences in the prevalence and characteristics of LLJs produced by the different WRF model ensemble members and reanalysis. The ERA5 captures too few LLJs (average underestimation of about 80 % at the sites), making meaningful statistical comparisons difficult, however, the results show that ERA5 does not capture the annual and diurnal cycles well, placed the LLJs too high up, and does not resolve the shear and veer structure when compared to the observations. However, when a larger detection window up to 500 m is used, the large-scale relative spatial distribution of mean LLJ occurrence rates shows largely similar characteristics as the WRF ensemble members, suggesting that the relative

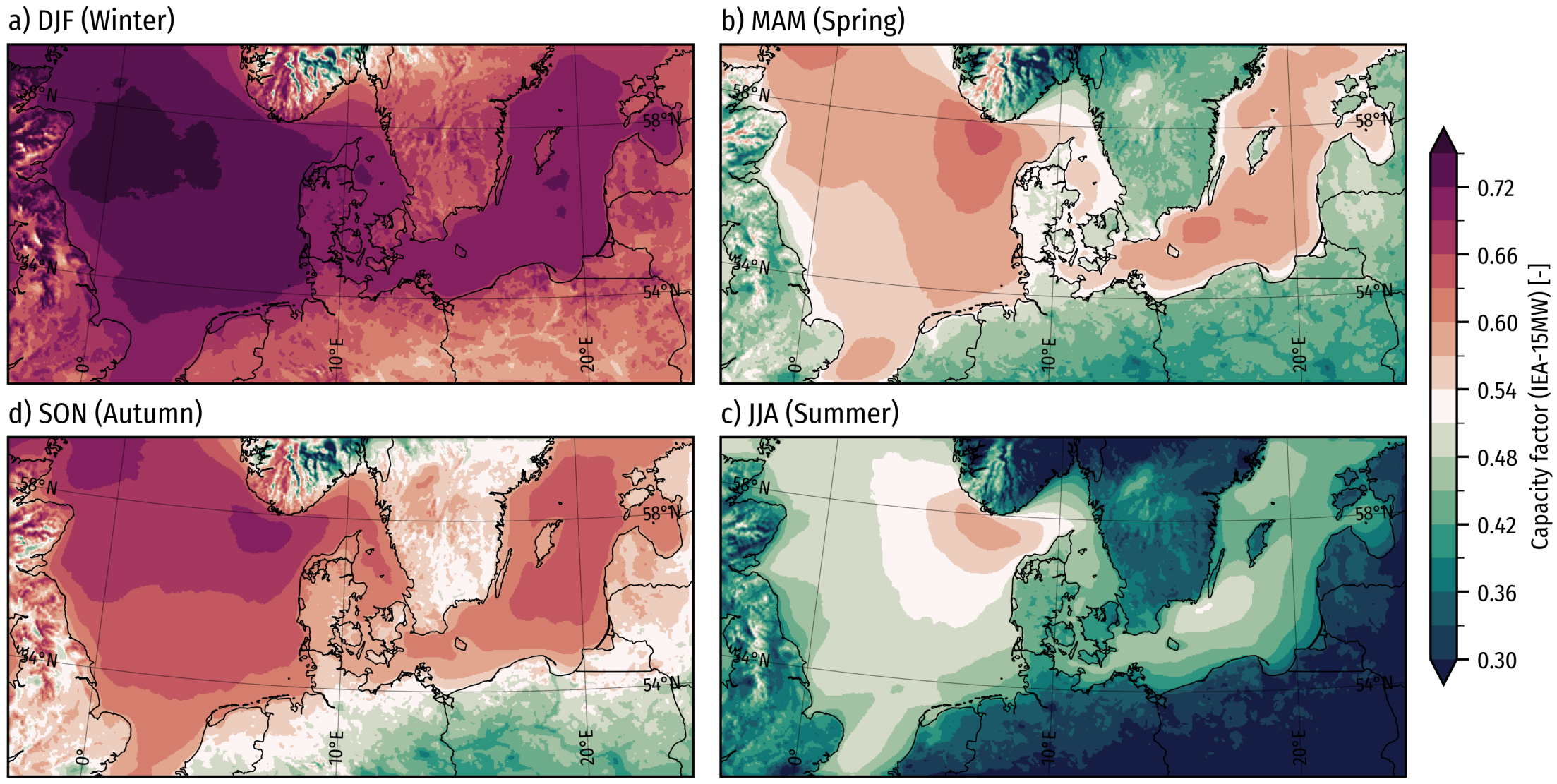

**Figure 19.** IEA 15MW capacity factor in the WRF model climatology (June 2019 to June 2024) separated by season.

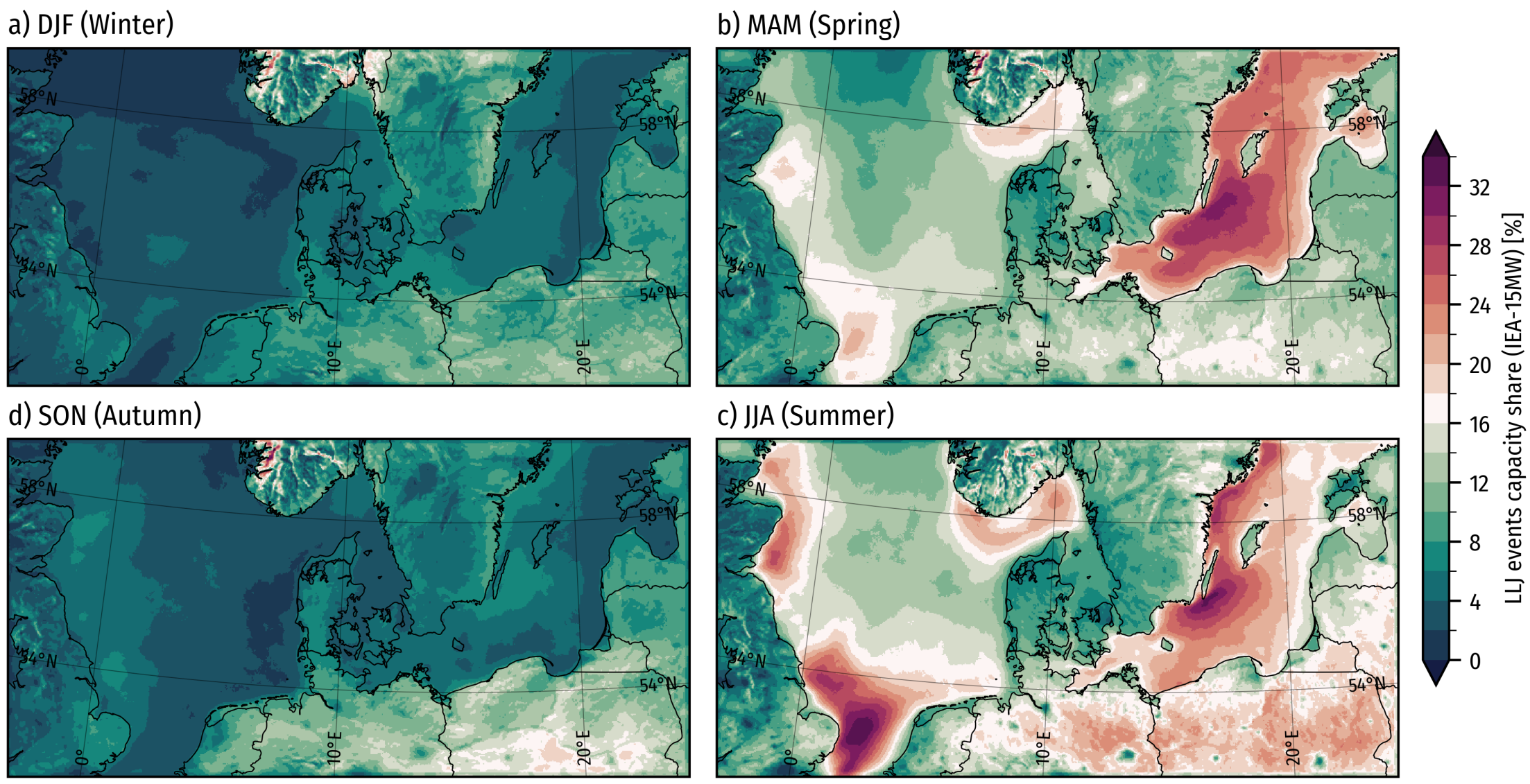

**Figure 20.** IEA 15MW capacity share happening during LLJ events in the WRF model climatology (June 2019 to June 2024) separated by season.

450 spatial distribution of LLJs is well captured in the ERA5. These findings agree with Kalverla et al. (2019), who assessed ERA5's ability to capture LLJs in the North Sea, showing that ERA5 tends to smear out the LLJs and place them too high up, but capturing a lot of the relative spatial-temporal variation. In their study, similarly to ours, higher LLJ rate-of-occurrences offshore in ERA5 were observed particularly near the Southern Bight, the UK east coast, and around Skagerrak.

Large differences in the simulation of LLJs also exist between the WRF ensemble members. At the five sites, using the MYJ PBL scheme (E_MYJ) results in an overestimation in the observed LLJ rates by 30 % on average, while using the MYNN 2.5 PBL scheme with the mixing length option "bl_mynn_mixlength=0" (E_MYNN_ML0) results in an underestimation by more than 70 %, using "bl_mynn_mixlength=2" (E_MYNN) results in a mean LLJ rate bias of around 15 %. In both cases, the mismatches made by E_MYJ and E_MYNN_ML0 are stronger offshore, while LLJ rates are generally more similar between the models onshore. The causes for higher rates of LLJs using MYJ deserve further investigation. A possible explanation is an overestimation of atmospheric stability of the stable marine boundary layer. Studies have indicated an overestimation of stable boundary layers onshore in some cases (Tastula et al., 2015). The E_MYNN_ML0 was included in the ensemble because the setup closely resembles that used for NEWA (Hahmann et al., 2020; Dörenkämper et al., 2020), "bl_mynn_mixlength=2" (E_MYNN) has become the default option and was implemented to improve the turbulent mixing length formulation for stable boundary layers (Olson et al., 2019), which our study also suggests it has. Increasing the grid spacing from 3 to 1 km does not show any significant improvement to LLJ accuracy, nor does the increase of vertical levels from 85 to 125 when using the MYNN scheme.

It is interesting to note that when choosing a WRF model configuration, standard evaluation metrics will select a different model configuration from what will be chosen when optimizing for LLJs occurrence and characteristics. The reasons for this are multiple and might be related not only to how the PBL scheme's momentum in the vertical (Draxl et al., 2014) but also to the simulation of the interaction between the atmosphere and the surface where the conditions for LLJ formation develop. The investigation of such effects is unfortunately outside the scope of this paper.

The LLJ climatology highlights regions favorable to LLJ occurrence, especially in the Baltic Sea and along the UK east coast, the straight of Dover, and the Dutch and German seas in spring and summer. As previously discussed, the spring and summertime LLJs in the Baltic Sea have been explained by the strong thermal contrasts between warm air over land and the colder sea surfaces persisting in those seasons Smedman et al. (1996). This causes frictional decoupling at the coastline and results in internal oscillation in space and the occurrence of a super-geostrophic jet (Blackadar, 1957; Smedman et al., 1995). Smedman et al. (1996) suggest that, with increasing travel time over the sea, the LLJs can sometimes transition from the "Blackadar"-kind to inversion-capped LLJs in a neutral well-mixed boundary layer. The formation at the coastline and gradual transition with travel time over the sea may explain the elevated rates close to the coastlines and the tendency for LLJs to increase in height and "depth" with distance to the coast. The most prevalent hours in LLJ climatology (Fig. 15c) are also consistent with this, as the occurrence peaks happen in late afternoon hours, but are gradually delayed with distance to shore. The strong thermal contrast in the spring and summer suggests some low-level baroclinicity and sea breezes also play a role. So-called "*corkscrew*" sea-breezes have a strong coastline-parallel component with land to the left in the Northern hemisphere (Miller et al., 2003), implying higher pressure over the sea and lower over land. *Corkscrew* sea breezes are associated with coastal jets (Steele et al., 2015), making these situations more likely to occur in our LLJ detection algorithm. The averaged LLJ wind directions in the LLJ climatology show strong coastline-parallel components in wind directions in spring and summer, which could indicate the influence of *corkscrew* sea breezes. The Dover strait jet (Capon, 2003; Hunt et al., 2004; Steele et al., 2015) shows up clearly in the climatology with peak occurrence in the summer. The most likely driving mechanism for the

jet is land-sea roughness and temperature contrasts, the first causing convergence over land when the wind coming from the sea decelerates over higher surface roughness, resulting in an internal boundary layer and accelerated recirculation winds and subsidence over the sea, and the latter producing *Corkscrew* sea-breezes (Capon, 2003; Hunt et al., 2004; Steele et al., 2015). Orographic channeling is not expected to play a major role (Capon, 2003), but there are some indications around the southern tip of Norway. In the LLJ climatology that we present here, the longest-lasting jets are found near the strait of Dover in the summer, which is consistent with is more strongly tied to synoptic scale weather conditions and wind directions than other coastal jets, e.g. in the Baltic Sea, which may be tied more strongly to the diurnal cycle.

One key challenge in LLJ studies is obtaining comparable LLJ rates across studies due to variations in spatio-temporal resolution and LLJ detection methods. Measurement resolution is constrained by instrument accuracy, averaging volume (in the case of remote-sensing), and sampling rates, while reanalysis and mesoscale models are limited by their resolvable scales and post-processing methods, e.g. interpolation and resampling. The spatio-temporal resolution acts as an implicit filter for LLJs, while the detection method serves as an explicit filter. Both directly influence the characteristics of detected LLJs. Additionally, the vertical window used for detection significantly affects which LLJ samples are retained (Kalverla et al., 2019). In this study, the simulated LLJ climatology, using the best-evaluated WRF model configuration, shows LLJ rates of ≈12 % at the Baltic measurement sites, and ≈6 % for the North Sea and Østerild N. sites, using all model levels up 500 m. These rates are much greater than the observed rates (0.5 % to 2.1 %) due to the different windows used. See Appendix B) for more on this. Consequently, we have mainly focused on relative rates and spatial patterns, rather than absolute rates, when presenting the climatology and comparing it to other studies.

## 7 Summary and conclusion

Offshore LLJs are complex flow features that represent an important wind resource, especially in specific parts of the North and Baltic seas. Understanding not just how they boost power generation, but also their impact on wind farm reliability and wake losses via modulation of turbulence and wake dissipation remains an important area of study for future wind farm planning and development.

Herein, we have presented a validated high-resolution mesoscale LLJ climatology of the North and southern Baltic Seas based on a five-year simulation using an optimized configuration of the WRF model. The climatology is openly available (Olsen et al., 2024). To choose the best WRF configuration for LLJ modeling in the region we evaluated nine different configurations, varying the PBL scheme, the vertical levels, horizontal grid spacing, and grid nudging. The evaluation is done by comparing the models against observations from FLSs and a mast at five sites, three in and near the North Sea, and two in the Baltic Sea. The evaluation period is 70 individual days distributed throughout one year from December 2021 to December 2022. We use a modified version of the Baas et al. (2009) method for LLJ detection. In the measurements, we detect about three times more LLJs in the Baltic Sea sites than in the North Sea area. Spring and summer are the most prevalent seasons and nighttime, early morning, and afternoon hours were the most prevalent hours for the Østerild, NS1-2, and BS1-2 sites respectively. LLJs lasted at most a few hours, with most only showing up in one one-hour sample. The mean height of detected LLJs varied from 104 m

to 121 m, but with many jets below 100 m. Wind during LLJ events comes mostly from easterly and southerly directions at NS1-2 and BS1-2 and from west and east at Østerild N.

Our model evaluation shows that using the ERA5 reanalysis for LLJ characterization is insufficient, hence the need for downscaling with the WRF model to generate the climatology. While several WRF ensemble members capture LLJ characteristics well at the sites, the member using the 3DTKE PBL scheme is ultimately chosen for the climatology because it captures most aspects of LLJ characteristics and occurrence rates well.

Our LLJ climatology highlights well-known areas favorable to LLJ occurrence. Particularly the Baltic Sea, along the UK east coast, in the Strait of Dover, and Skagerrak. The novelty is the validated climatology covering a wide area used for wind energy development and the number wide range of characteristics made available that help indicate the spatial variations in LLJ rates, seasonality, heights, durations, direction, levels, and heights, and magnitude of maximum shear.

*Code availability.* The WRF model is freely available via https://github.com/wrf-model/WRF. The code used for data processing and analysis is available upon request.

*Data availability.* The LLJ climatological dataset is freely available (Olsen et al., 2024). The Floating LiDAR system measurements were made available by EnergiNet. The ERA5 reanalysis data is available from the Copernicus Climate Change Service (C3S): Complete ERA5 global atmospheric reanalysis. DOI: 10.24381/cds.143582cf (Accessed on 22 October 2024). The NEWA mesoscale data can be downloaded from https://map.neweuropeanwindatlas.eu/. Freely available base maps from https://www.naturalearthdata.com/downloads/ were used for maps.

**Table A1.** Stability classification based on Obukhov length adapted from Gryning et al. (2007)

| Class | Obuhkov length (L) |
|---|---|
| Very stable (vs) | $L \leq 50$ |
| Stable (s) | $50 < L \leq 200$ |
| Near-neutral stable (nns) | $200 < L \leq 500$ |
| Neutral (n) | $L > 500$, $L < -500$ |
| Near-neutral unstable (nnu) | $-500 \leq L < -200$ |
| Unstable (u) | $-200 \leq L < -100$ |
| Very unstable (vu) | $-100 \leq L$ |

## Appendix A: Stability during observed LLJ observations

In our study, we present characteristics of LLJs in the observations. Here we further provide some evidence for the atmospheric conditions during LLJ events in our samples. Figure A1 shows distributions of hub-height wind speeds, air-sea temperature differences (at the offshore sites), and atmospheric stability classifications for all conditions and LLJ events. ERA5 data (nearest in time and space) provided temperature differences and stability classes, while wind speeds were measured directly. The stability classes are adapted from Gryning et al. (2007) and shown in table A1.

The figure shows that LLJs typically occur at intermediate wind speeds, avoiding both the weakest and strongest winds. Offshore, LLJs are linked to positive air-sea temperature differences (warm air over cold water) and stable atmospheric stratification. At Østerild N., LLJs also prefer stable conditions, though they can occur under neutral and unstable conditions as well. The strong tendency for LLJs to occur during stable conditions is evidence that frictional decoupling is the likely mechanism of formation, resulting in inertial oscillations in time or in space with a period of super-geostrophic winds.

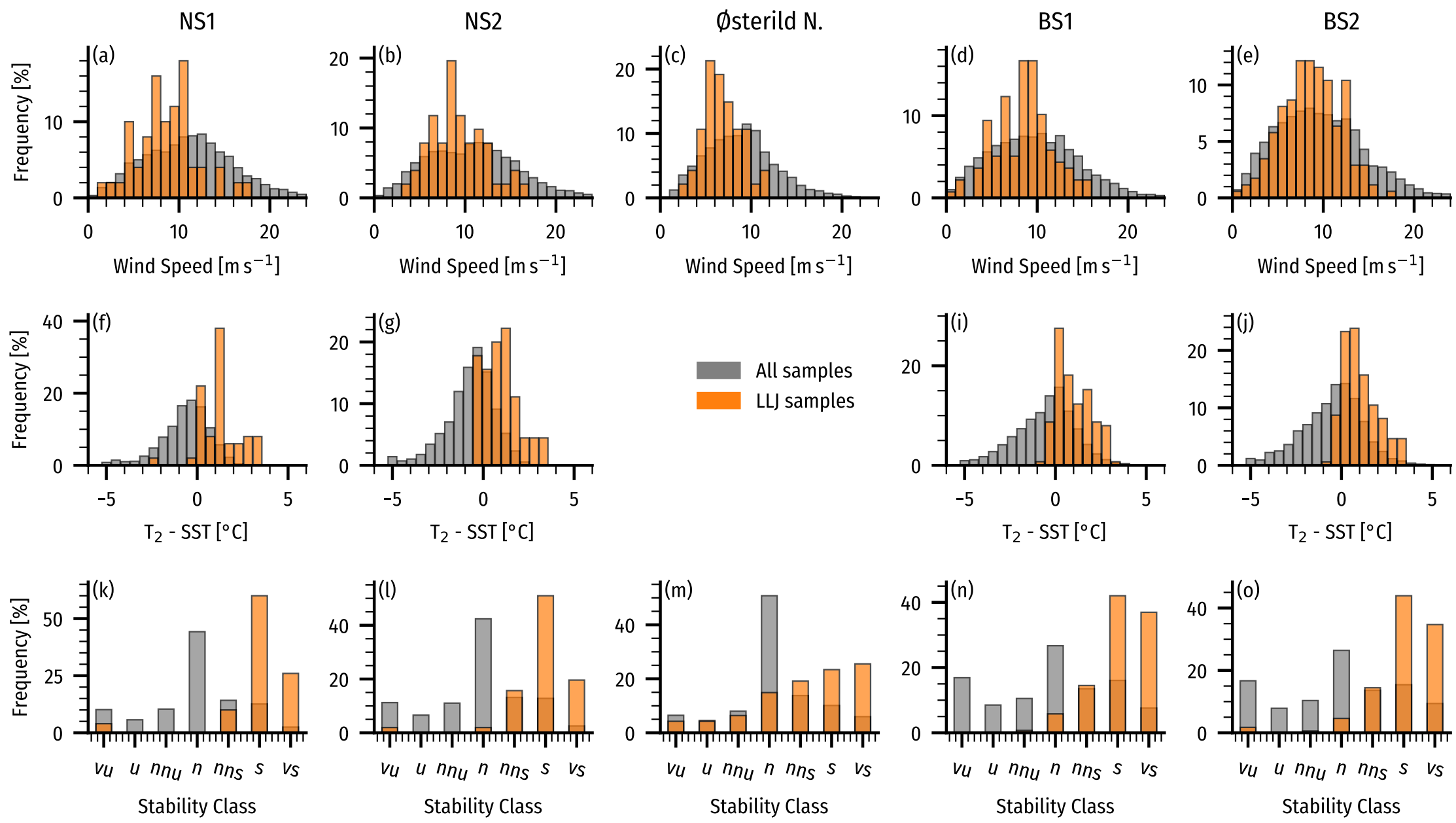

**Figure A1.** Frequency distributions of wind speeds at 150 m (top row), air-sea temperature difference (middle row), and atmospheric stability (bottom row) at the five sites. The two distributions are for all samples (grey) and using the LLJ samples (orange). The air-sea temperature difference and the stability category come from the co-incident (nearest in time and space to the measurements) ERA5 reanalysis data.

## Appendix B: The sensitivity to the maximum height for LLJ detection

Expanding the vertical window for LLJ detection results in more jets being detected. To understand the sensitivity to the maximum height of detection, $z_{\max}$ in Fig. 2 of the main text, we repeatedly detected LLJs in the wind speed profiles of the observations and evaluated models for the 70-day evaluation period, varying $z_{\max}$ from near the surface resulting in no LLJs being detected up to 500 m which was the height we aimed to use for the climatology, the observations only goes up to 270 and 244 m so they are capped there. We used the native levels of each model but discarded any levels below 25 meters to better align the lower height-bound with the measurements.

Figure B1 displays the number of jets detected as $z_{\max}$ increased. It shows the strong sensitivity to the vertical window for the rate of LLJ detection. Taking just E_MYJ at BS2 for example, the rate is about 4 % at 200 m but more than 10 % already at 300 m and grows to more than 20 % somewhere between 400 m to 500 m. E_MYJ does have the steepest increase, but the measurements and other models also show a rapid increase with $z_{\max}$. The figure also reveals significant discrepancies among the models. ERA5 and the E_MYNN_ML0 model detect considerably fewer LLJs compared to the others, whereas E_MYJ detects substantially more, with a sharp increase in LLJ numbers as $z_{\max}$ rises, surpassing the increase indicated by the measurements. Overall, the E_3DTKE and E_MYNN models most closely align with the LLJ rates observed in the measurements. A minor increase in the number of LLJs is observed across most models when the grid spacing is reduced from 3 km to 1 km. However, the impact of changes in PBL schemes and the number of vertical levels is more pronounced than the effect of horizontal resolution in this context.

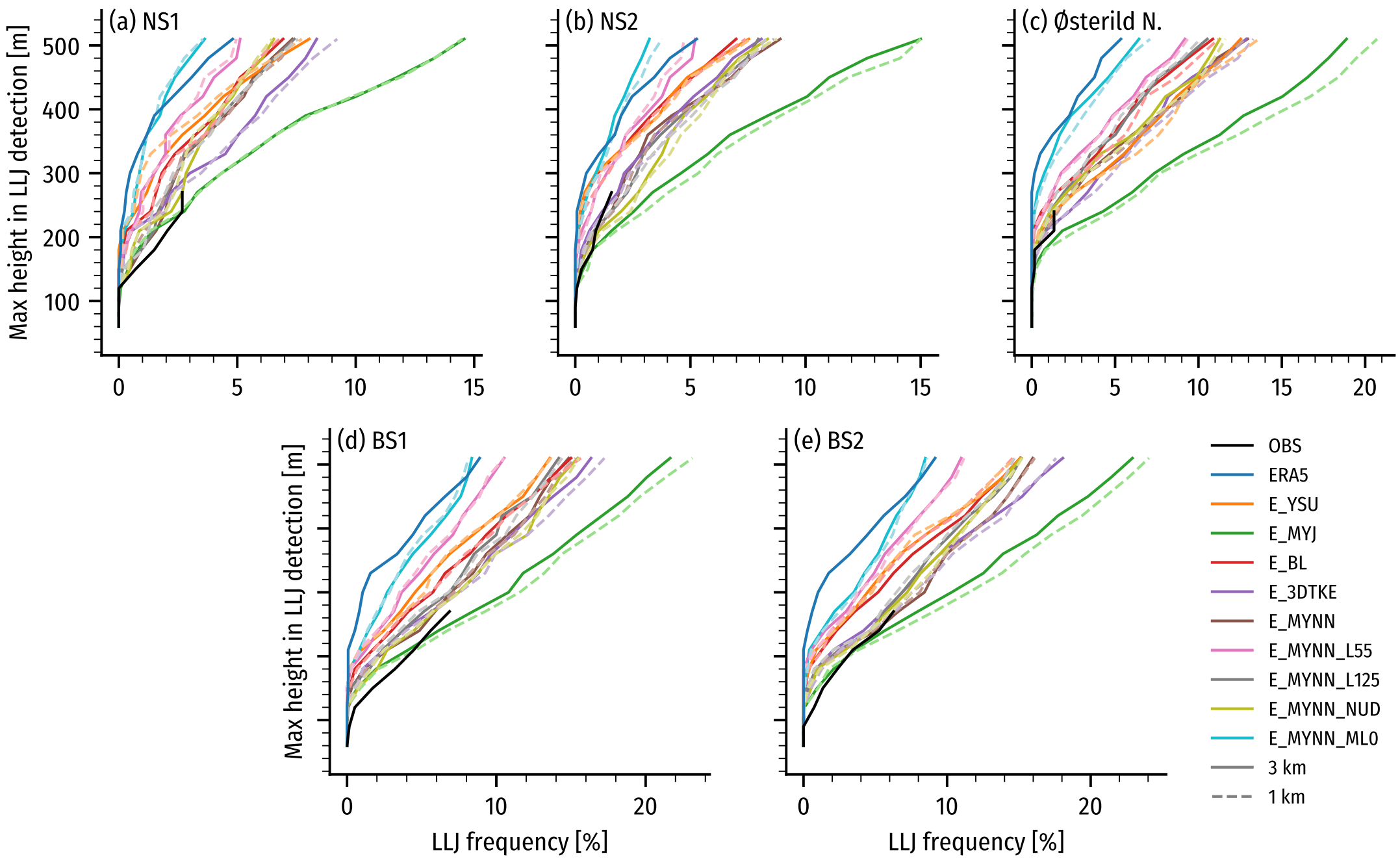

**Figure B1.** Rates of LLJ detection with increasing maximum height used in detection using the native height levels of each observational or modeled dataset. For the modeled datasets, to bring the available vertical levels more in line with the observations, only levels above a height of 25 m were used here

**Table C1.** LLJ detection thresholds, weak, moderate, and strong. Each threshold is defined based on the minimum fall-off both above and below the LLJ core, both in absolute and relative terms.

| Threshold | Fall-off | |
|---|---|---|
| | Absolute | Relative |
| Weak | 1.0 m s$^{-1}$ | 10 % |
| Moderate | 1.5 m s$^{-1}$ | 15 % |
| Strong | 2.0 m s$^{-1}$ | 20 % |

## Appendix C: Evaluating LLJ detection thresholds

As described in section 2.4, the selection of LLJ detection thresholds varies across studies, and there is no strong consensus. To understand the sensitivities to the thresholds in our study, we performed a sensitivity analysis with three levels of thresholds termed "weak", "moderate", and "strong" detection thresholds (Table C1). The moderate thresholds are our chosen thresholds in the main part of this paper. We evaluated the thresholds for the model ensemble, for the same 70-day period as in Section 4, focusing on spatial Z-score-difference distributions between two threshold levels for the LLJ occurrence rates. We chose to focus on these differences because they indicate how stable the spatial variation of LLJ occurrence rates is to changing thresholds, providing a good indication of the generality of the spatial variations of our final WRF LLJ climatology. We analyzed the Z-score difference histograms (Figure C1 and standard deviation ($\sigma_{Z_{\mathrm{LLJ}}}$) maps (Figure C2) to define how sensitive a given model ensemble is to the threshold selection and what geographical areas are most sensitive. If the histogram kurtosis is higher (narrow distribution), the spatial variation of LLJ rates is robust to changes in the LLJ detection threshold. On the other hand, if the histogram displays greater width (lower kurtosis), there will be larger differences in spatial variation with different thresholds used.

We refer to the baseline thresholds from Baas et al. (2009) as "strong", and analyze the relative differences in spatial variation to more moderate threshold levels that expand the subset of LLJ events to obtain more statistically robust results.

Figure C1 shows the Z-score-difference histograms for every model ensemble plus the NEWA and ERA5 datasets. Notably, the "moderate-strong" and "weak-moderate" histograms are very similar, indicating that similar differences arise for these two similar jumps in threshold magnitudes (0.5 m s$^{-1}$ and 5 % ). The "weak-strong" histograms are wider, showing that the spatial distribution of variations continues to change with threshold magnitude. The ensemble-member-specific results show that some members (e.g. all the MYNN-based members) have narrower distributions, indicating robustness to thresholds, while other members have wider distributions (E_MYJ, E_3DTKE, E_BL, and ERA5), indicating larger reconfiguration in the spatial variation with changing thresholds. The maps of $\sigma_{Z_{\mathrm{LLJ}}}$ in Fig. C2 show that the differences manifest in largely ensemble-member-specific locations, but tend to be around complex terrain (Norwegian mountains), where higher rates are

produced (Baltic Sea and near Hamburg), and where the largest gradients of LLJ occurrence are in Fig. 11. The biggest differences are seen in the BL and ERA5 panels.

In conclusion, the spatial variation in the rate of LLJ occurrence is sensitive to the thresholds and model-specific sensitivities should be expected.

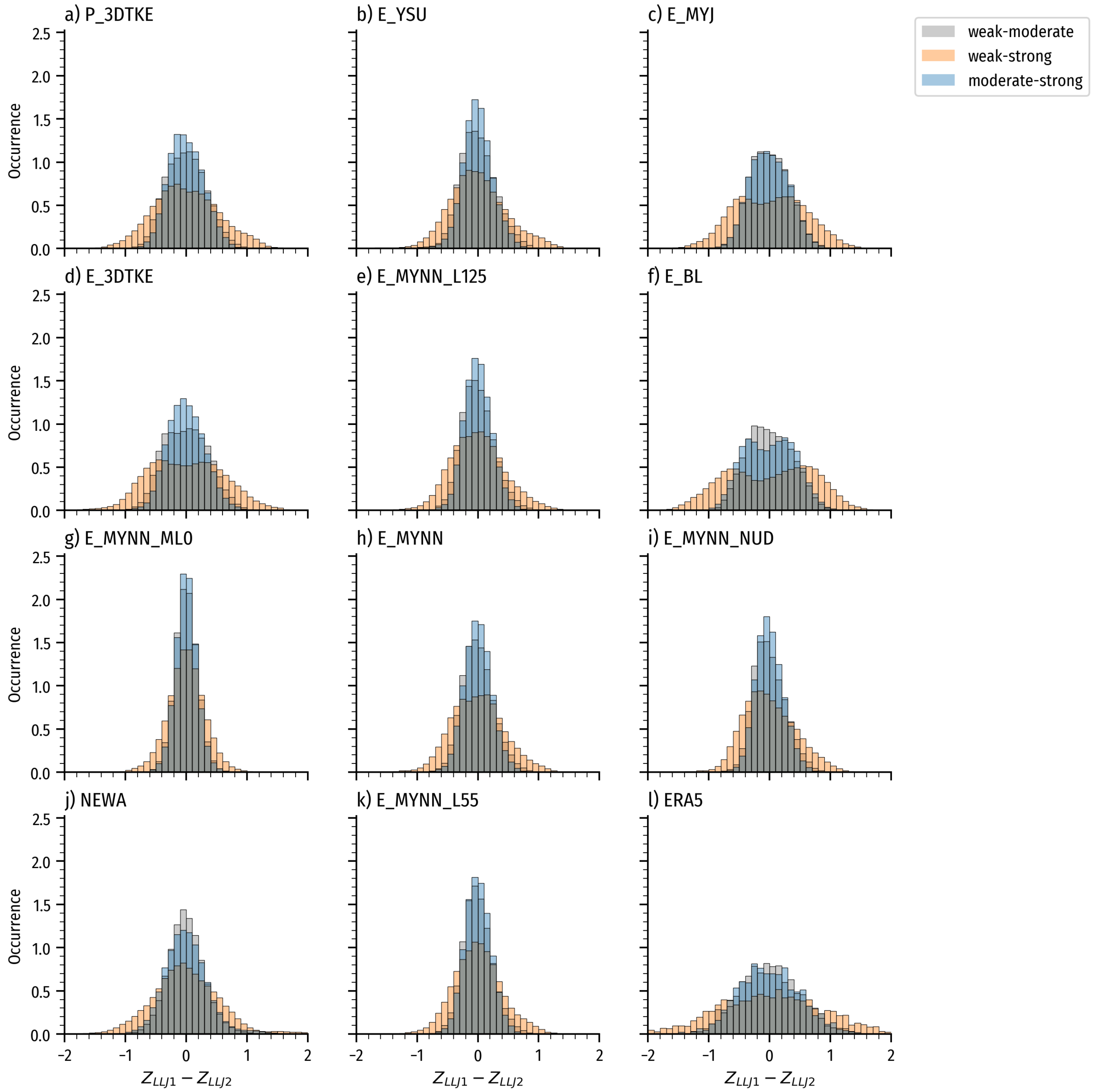

**Figure C1.** Z-score-difference histograms for three LLJ detection thresholds. The gray histograms show the weak-moderate differences, the orange histograms show the weak-strong differences, and the blue shows the moderate-strong differences. Panels from a to j contain the WRF ensembles and k and l contain the NEWA and ERA5 datasets, respectively.

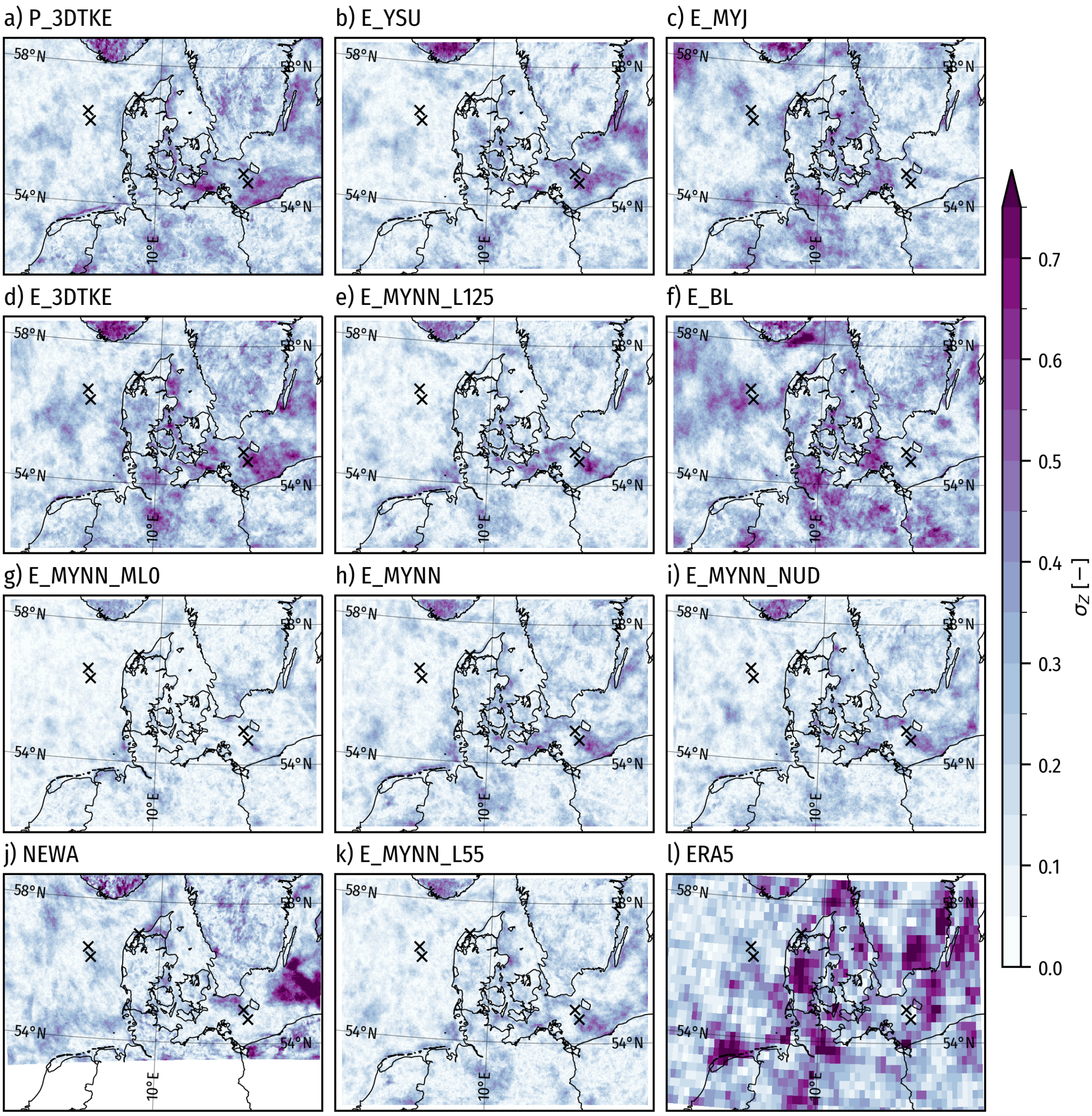

**Figure C2.** Z-score standard deviation for three LLJ detection thresholds. (a)–(i),(k) WRF ensemble members in Table 2, (j) NEWA, and (l) ERA5. The domains are slightly different for NEWA and ERA5 compared to the WRF domain used by the ensemble members. The black markers show the measurement sites.

**Table D1.** Number of simulation days per month in selected sample for ensemble evaluation

| Month | Jan | Feb | Mar | Apr | May | Jun | Jul | Aug | Sep | Oct | Nov | Dec | Total |
|---|---|---|---|---|---|---|---|---|---|---|---|---|---|
| LLJ | 0 | 0 | 9 | 5 | 7 | 6 | 4 | 7 | 0 | 2 | 1 | 6 | 47 |
| Extra | 5 | 5 | 0 | 0 | 0 | 0 | 0 | 1 | 0 | 5 | 3 | 4 | 23 |
| Total | 5 | 5 | 9 | 5 | 7 | 6 | 5 | 7 | 5 | 5 | 5 | 6 | 70 |

## Appendix D: Simulation days

The 70 days used for model evaluation were chosen based on two criteria: first, 47 days were selected because LLJs (at least $2\,\mathrm{m\,s^{-1}}$ and 20 % winds peed fall-offs between the maximum and the minima) occurred in the measurements at one or more of the sites during that day. Second, to balance the annual distribution 23 additional days were selected randomly, but stratified by month, to obtain at least five days per month. Table D1 shows the number of days chosen based on each criterion.

## Appendix E: Additional insights from LLJ climatology

### E1 Wind shear and veer above and below the jet core

Here we present additional layers from the LLJ climatology to show the spatial variations in the mean shear and veer above and below the LLJs. Figures E1 and E2), show maps of seasonal mean values of the maximum (absolute) wind speed shear ($\Delta U/\Delta z$) found between the LLJ core and the wind speed minimum below, and the minimum (largest negative value) found between the core and the wind speed minimum above. Colors are scaled to show the variation for offshore regions. Onshore, the shear below the jet cores is dominated by the higher roughness relative to the sea. The maps show that winter and spring are especially associated with stronger shear over the Baltic Sea, while it tends to be slightly weaker in summer and fall. In the North Sea, the strongest shear below the jets occurs in winter and near geographic features and coastlines year-round. Above the jets, the strongest negative shear happens in spring and summer, particularly close to the UK east coast, the Netherlands, and south of Norway.

Figures E3 and E4 show the mean veer between the LLJ core and the wind speed minima below and above respectively. The veer maps show similar trends as the shear shown above, with a higher mean veer below the jet core in spring and summer. For the veer above the jet core, the seasons and locations with stronger wind speed shear are associated with weaker mean veer, reducing the negative veer (backing) that is present most of the time. Thus, offshore LLJs are associated with a reduced veer above the jet core.

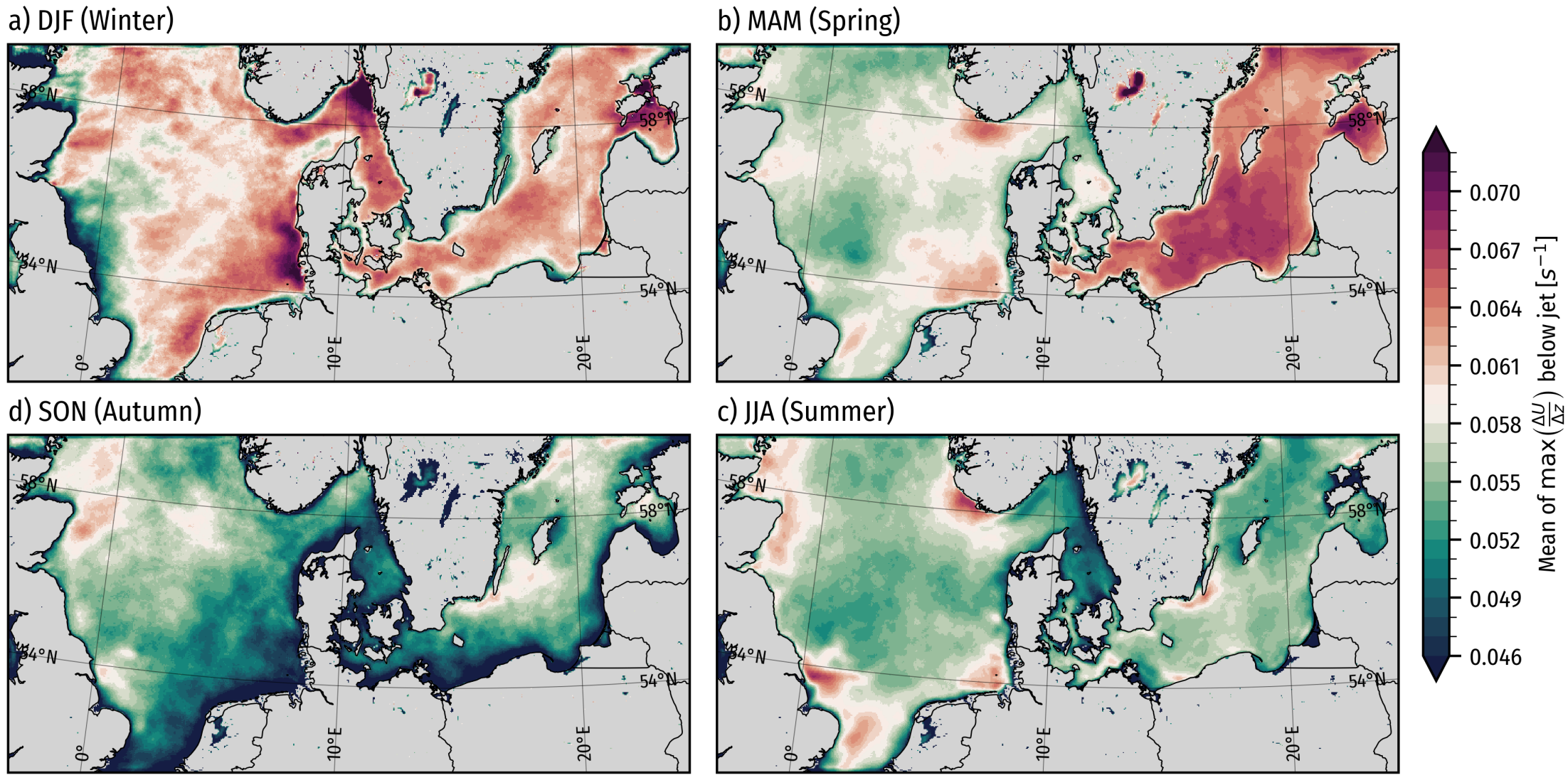

**Figure E1.** Mean of max $\left(\frac{\Delta U}{\Delta z}\right)$ in the WRF model climatology (June 2019 to June 2024) separated by season. Onshore, the shear is dominated by surface roughness effects, so these are greyed out to keep the focus on offshore locations.

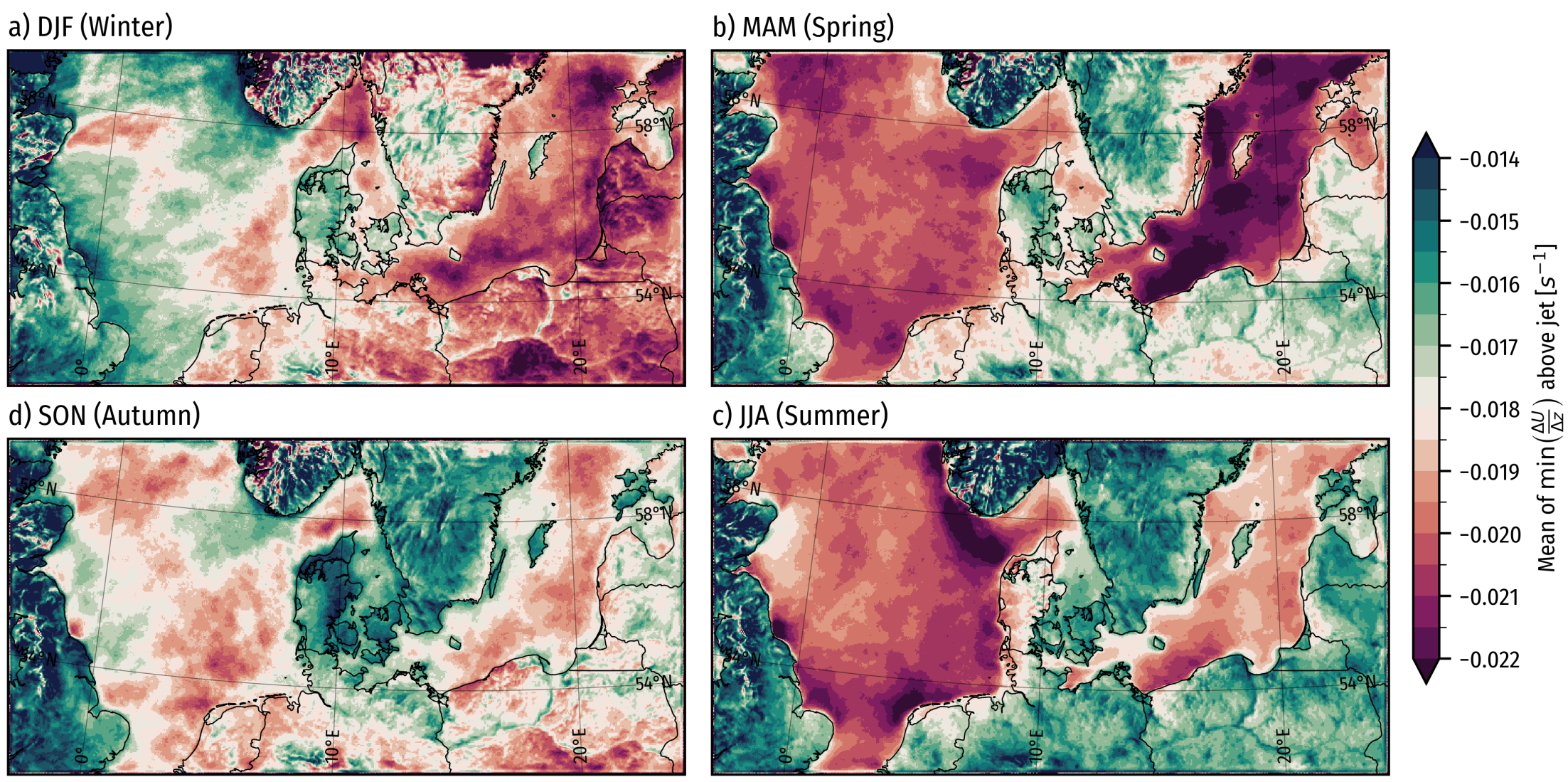

**Figure E2.** Mean of min $\left(\frac{\Delta U}{\Delta z}\right)$ in the WRF model climatology (June 2019 to June 2024) separated by season.

## E2 Distributions of LLJ characteristics in climatology at three offshore sites

The LLJ climatology presented shows large spatial and seasonal variations in LLJ occurrence rates, jet heights, jet duration, shear, and veer. To provide further insights, we present the distributions of several key LLJ characteristics for three offshore sites. The three sites, shown in Fig. E5 were selected due to their representativeness for some of the more extreme locations in

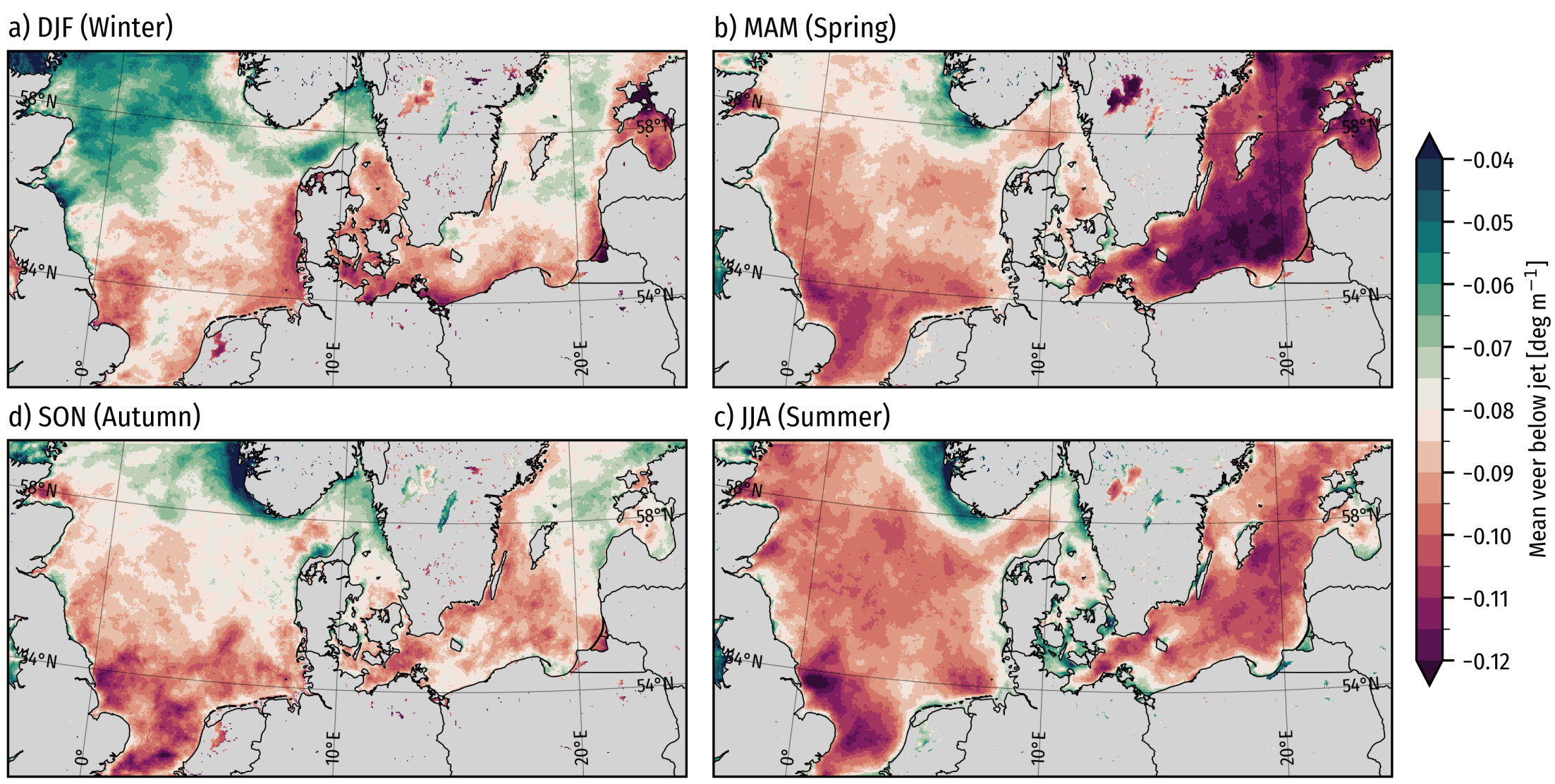

**Figure E3.** Maps showing the mean in time of the mean veer between the wind speed minimum below the jet and the jet core in the WRF model climatology (June 2019 to June 2024) separated by season. Onshore, the veer is dominated by surface roughness effects, so these are greyed out to keep the focus on offshore locations.

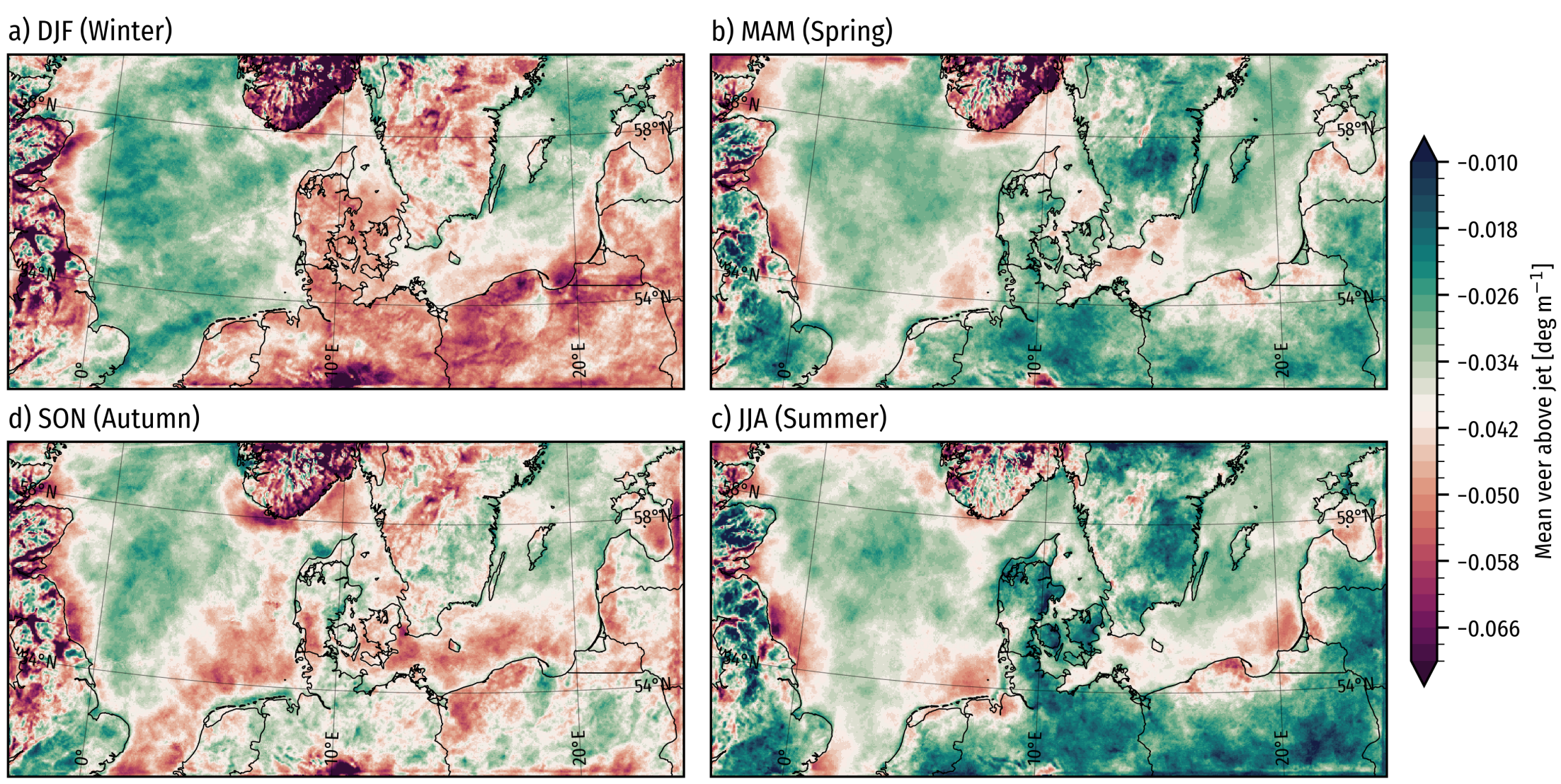

**Figure E4.** Maps showing the mean in time of the mean veer between the jet core and the wind speed minimum above in the WRF model climatology (June 2019 to June 2024) separated by season.

one aspect or another, related to LLJs. Öland South was selected for the high LLJ occurrence rate and to represent the Baltic

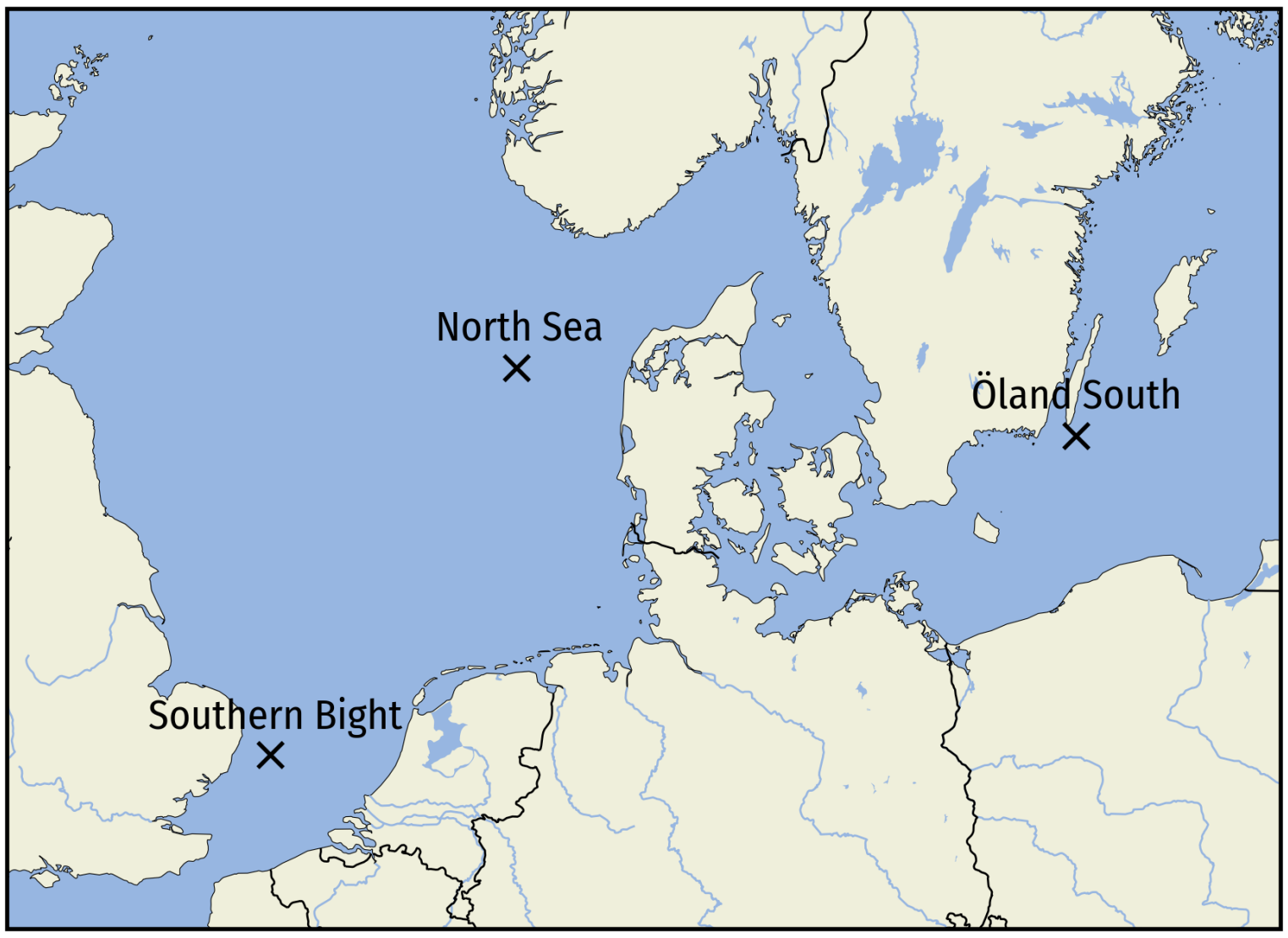

**Figure E5.** Locations of the three chosen sites for showing distributions of selected LLJ characteristics.

Sea. Southern Bight was chosen for its high LLJ occurrence rate and long-lasting jets. Finally, The North Sea site was selected to serve as a low-occurrence reference site in an active development region.

Figure E6 shows distributions of shear and veer, above, and below, the jet core, as well as the jet height, duration, and wind speed. The figure shows that, although all three sites have similar distributions across all the metrics, some notable differences can be seen. In particular, LLJs in the North Sea are placed higher up (a few tens of meters across the distribution), have slightly weaker shear, and veer below the jet core, and jets tend to last for less time, compared to the two other locations. The two sites, Öland South and Southern Bight have more similar distributions (compared with the North Sea), but jets last longer at Southern Bight, showing more cases with very long-lasting jets.

*Author contributions.* ANH and BTO planned and scoped out the study, with useful suggestions from MD and MZ. ANH set up the WRF model configuration and conducted the simulations. BTO preprocessed measurements and model data for analysis. BTO and NGA carried out the model evaluation. BTO created the LLJ climatological layers from the long-term WRF model simulations. All authors participated in the writing, editing, and internal reviews of the manuscript.

*Competing interests.* The Authors declare no competing interests are present

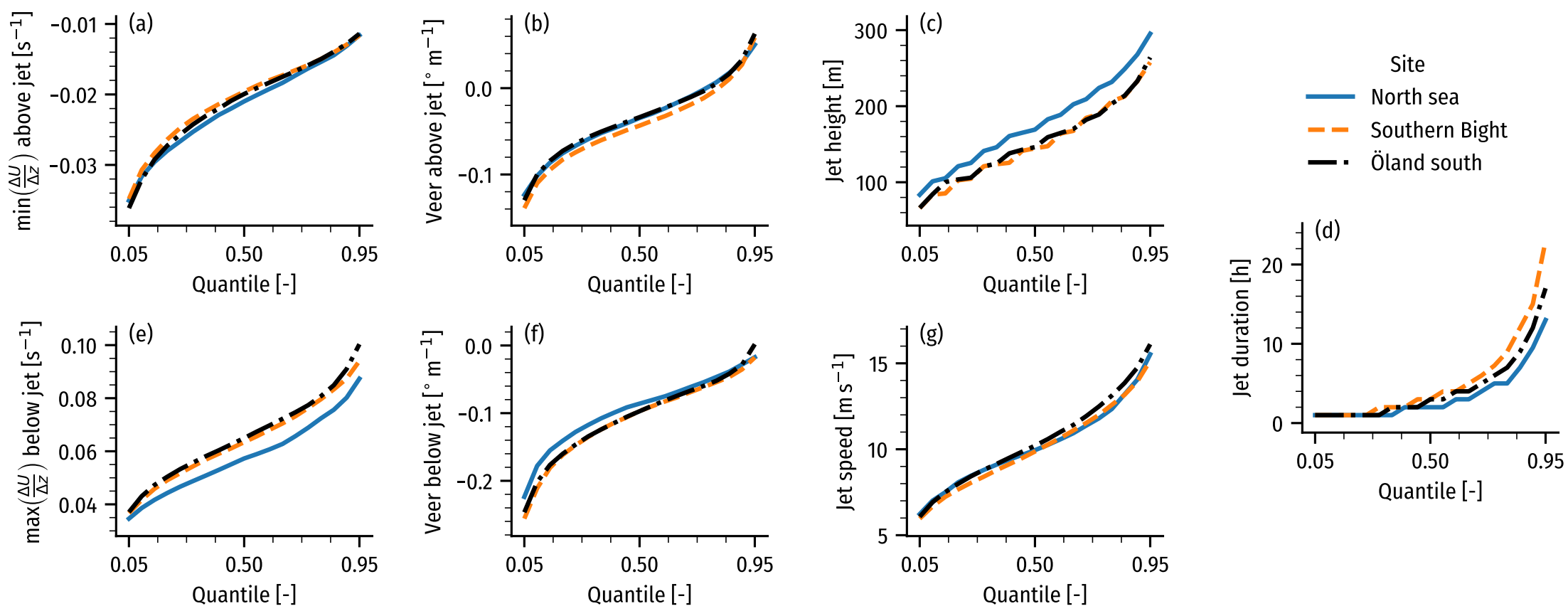

**Figure E6.** Distributions at the three chosen sites of: (a) min $\left(\frac{\Delta U}{\Delta z}\right)$ above the jet core, (b) veer between the jet core and the wind speed minimum above, (c) jet height, (d) jet duration, (e) max $\left(\frac{\Delta U}{\Delta z}\right)$ below the jet core, (f) veer between the jet core and the wind speed minimum below, (g) jet speed.

*Acknowledgements.* We express our sincere gratitude to EnergiNet for providing access to the floating LiDAR system datasets from the North Sea and Baltic Sea, and to DTU wind's measurement section for organizing the Østerild measurement data. This work was supported by the European Union Horizon Europe Framework Programme (HORIZON-CL5-2021-D3-03-04) under grant agreement no. 101084205. ANH was partly funded by the Independent Research Fund Denmark through the 'Multi-scale Atmospheric Modeling Above the Seas' (MAMAS) project (nr. 0217-00055B). We thank the European Center for Medium Range Forecast (ECMWF), its members, and the Copernicus Climate Change Service Climate Data Store (CDS) for making the ERA5 dataset openly available and the UK's Met Office for the availability of the OSTIA SST dataset. We acknowledge the use of the Sophia HPC cluster operated by DTU. We are thankful to open-source developers of maintainers for the open-source Python libraries used for data processing, analysis, and visualization: numpy (Harris et al., 2020), scipy (Virtanen et al., 2020), xarray (Hoyer and Hamman, 2017), numba (Lam et al., 2015), matplotlib (Hunter, 2007), cartopy (Met Office, 2010 - 2015), Python optimal transport (Flamary et al., 2021), and many others. We used large-language models, including ChatGPT, to assist in the editing of parts of this paper.

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
