# Peer review of "Low-level jets in the North and Baltic Seas: Mesoscale Model Sensitivity and Climatology using WRF V4.2.1"

_EGUsphere, 2024_

## Author Response (AR1)

**Low-level jets in the North and Baltic Seas: Mesoscale Model Sensitivity and Climatology**

Bjarke T. Olsen, Andrea N. Hahmann, Nicolas G. Alonso-de-Linaje, Mark Žagar, and Martin Dörenkämper

**Response to Comments by Reviewers**

We appreciate the insightful feedback from both reviewers. We have addressed all comments in detail below and believe the revisions have significantly enhanced the manuscript. We hope the reviewers and editor find the revised version suitable for publication.

Reviewer comments are in black, and our responses are in blue

**REVIEWER #1**

Summary: This paper studies the climatology of low-level jets (frequency, intensity, wind maximum height, duration) in observations over the North Sea and the Baltic Sea, and how these are represented in the WRF model, NEWA (wind atlas) and ERA5. For WRF an ensemble of configurations is set up in terms of PBL physics, number of model layers and domain size. It is found that the ERA5 model underestimates LLJ speed and dilutes the LLJ over a too deep layer. There appears to be considerable difference in timing, location and shapes of the LLJ climatology between WRF configurations. This is important information for wind energy resource assessment, and load assessment and for the development of future wind atlases.

Recommendation: revise

Major remarks:

1. The abstract is relatively long and reports quite some of the *activities* that were employed in the study. So now it is more like a summary. However, it would be good to rewrite the abstract such that it reports more about *what has been*

*learnt*, i.e. focus on the scientific knowledge/insights that were gained from the study.

Thank you for the feedback. We have rewritten the abstract accordingly, emphasizing what was learned more strongly.

2. I find the paper has quite many figures. On the one hand this means that the paper is very complete, but at the same time it is a less attractive invitation to read the paper. Also many of the figures are not discussed in very much detail (some in 4-5 lines). So it would be great if the paper can be revised such that the discussion is deepened (link model behaviour to atmospheric physics and dynamics, or to modelling strategies like nudging, initialization, time stepping etc etc), and end up with less figures.

We have moved one figure (Fig. 8) to its own Appendix and expanded the text discussion Fig. 4. While we agree that further analysis linking the model behavior to atmospheric physics and dynamics would be highly relevant (we have expanded the discussions around this topic in several places), given the length of the paper as it is, we deem this to be best suited for a future more focussed study, while this paper has a stronger focus on the LLJ climatology/atlas.

3. The members of the WRF ensembles do not have very intuitive names/labels, which means for any discussion of a specific member, the reader has to go back to the table where they have been defined. This is unhandy and time consuming. Could you perhaps do an effort to revise the names in a more intuitive way?

We agree that the names are not optimal. In our suggested changes we now talk about the three phases (phase one: sensitivity experiments, phase two: sensitivities towards production setup, and phase three: production) instead of "E," "S," and "P". We also renamed the "S" experiments to E_3DTKE-NUD, E_3DTKE-NUD-72H, and E_3DTKE-NUD-168H to hopefully make the progression more apparent from the E_3DTKE from stage 1 to the final production run. We propose to call the production member "P_CLIM". In the line of progression, it should probably be called P_3DTKE-NUD-168H, but P_CLIM seems more precise (more concise and signaling "production" and "climatology").

4. In terms of positioning the paper, one could link it more to earlier wind atlases results. In fact you make kind of a wind atlas with 5 y of WRF simulations. For a wind atlas it is a short time span, but the main sensitivities remain the same if you would have run 30 years. So I think the paper can benefit from being

positioned in a storyline of the making of wind atlases, and/or positioned in relation to recent wind atlas products like NEWA, DOWA, ... and whether similar biases are seen. For example Kalverla et al (2020, https://rmets.onlinelibrary.wiley.com/doi/full/10.1002/qj.3748) discusses similar diurnal and seasonal cycles of LLJ climatology in three wind atlases.

Good point. We have expanded on this in the introduction to the paper.

Minor remarks

Ln 6: we vary -> we perform sensitivity experiments....

Reformulated: " In the WRF model simulations, to assess model sensitivities, we vary the..."

Ln 8: replicate -> reproduce

Done. Thank you.

Ln 19: gives insights: what are these insights? Please add to the abstract.

We have revised the abstract to reflect these insights better.

Ln 32: During LLJ events, wind speeds increase, leading to -> LLJ events lead to

Done. Thank you.

Ln 37: I think that in addition to Kalverla et al. (2019), this is also a good reference based on observations over the North Sea: https://www.sciencedirect.com/science/article/pii/S0167610516307061

Thank you. We agree it's a valuable study to mention and have added it to the references.

Ln 67-69: These read as a figure caption. Better not to put in the main text.

We agree and have removed the introductory part of the Methods section, which duplicates the caption to Fig 1.

Ln 75: Please include some words whether the period is different or not from the typical climatology.

Excellent idea. We have offered some words on this in the section based on a comparison of the year 2022 to the period 2005-2024 in ERA5 at the North and Baltic Sea sites.

Ln 101: please add the model version of WRF

Done. Thank you.

Ln 115: if the goal of this final set of simulations is to realize a multi-year dataset, then P_3DTKE is not the most intuitive name to facilitate the reader.

We agree that the labeling is not optimal. We propose to use the label "P_CLIM".

Table 3: it is unclear how turbulent diffusion was dealt with. In the text and in Table the paper mentions a variety of PBL schemes, while table 3 suggests Smagorinsky closure was used everywhere. Please clarify.

Both are true; the PBL schemes handle the vertical mixing, diff_opt=2 enables horizontal mixing

Ln 143-144: change of the thresholds. Please justify more why this is allowed, and how much it affected your results. I.e. at least mention that the number of samples increased from XXX to YYY cases.

We have added this information. As we try to convey in the text, the definitions of a "low-level jet" are somewhat arbitrary, so we chose to dilute the thresholds to have more robustness in results.

Ln 152: ... consistent. Please add a sentence or two how your work is consistent in this context. How do you ensure you do the right thing for answering your research question.

We expanded on this by adding this paragraph:

"To ensure consistency between observations and models, we take several steps in this study. Observations are averaged into hourly means to better match the temporal variability of the mesoscale model and ERA5 reanalysis. Additionally, model outputs are interpolated to the measurement sampling heights for direct comparison. However, complete consistency is inherently unattainable due to differences in spatial scales:

point observations (or small volume averages, such as with the FLSs) are contrasted with models and reanalysis data that operate at much coarser spatial resolutions. This limitation, however, is a deliberate part of the study's aim, as we evaluate the model's ability to reproduce LLJs despite its resolution constraints"

Ln 160: log-linear interpolation. This will not work in the case of a present LLJ as you show in Fig 2. In which % of the cases do you interpolate wind speed profiles with LLJ properties using this method?

We use log-linear interpolation in height for all instances where we compare the models with observations. We interpolate the model-level data (variable height) from WRF or ERA5 to the (constant) heights of the observations. You can see the model levels in Fig. 3. Notably, the interpolation is done log-linearly in height between the model layers (not, e.g., from the surface and up), so the interpolated data remains constrained by the model levels, meaning low-level jet profiles (e.g., as in Fig. 2 can be well resolved). Choosing the proper interpolation method for a highly undulating profile, such as the case of LLJs, is difficult. Still, our approach does not meaningfully (negatively) impact the study's conclusions over other methods, such as linear-in-height interpolation.

Ln 172: This EMD method sounds interesting, but is not used very often (but that does not mean I am not confident in it). Most studies would likely use a Mann-Whitney-U test to test whether the distributions are the same or not. So why is this alternative EMD method preferred over more often used methods? Likely you have a good reason for it, and could be sold as an innovative part of the study.

The difference is that EMD is continuous and measures the statistical distance rather than testing if two samples are drawn from identical distributions. It considers the distance between samples, not, e.g., the divergence between distributions as in Kullback-Lieibler (KL) divergence. Two non-overlapping distributions would have KL=0 but continue to increase in EMD if pulled further apart.

Ln 182: shear estimation: please add a description how you include negative shear in your estimation? i.e. if you have 1 model layer with du/dz=1 and the layer above du/dz=-1 (like for a LLJ top), is the total shear then 0, or 2?

We use the instantaneous shear between two levels du/dz (using the halfway height between the levels to set the level of that shear). We thus have samples of du/dz (going

from some negative value to some positive value) at each half-way height from the observations and from the models. These shear distributions are then compared (using EMD) between the model and the observations at each height EMD(z). We then average across height to obtain a mean(EMD). EMD is, by its nature, unsigned, so we don't know if errors are generally caused by a negative or positive bias at any given level. Since we average across height, we also don't know what height errors are coming from. However, since we are looking for the best model to capture the spatial characteristics and ability to reproduce distributions, we believe this is a good evaluation metric to find the best model

Ln 190: Is it a problem for the Z score that the wind speed distribution is not normally distributed?

Since the Z-scores are not used for wind speed distributions, only for relative spatial variability of LLJ rates, which are closer to Gaussian, we believe it is appropriate to use Z-scores in this context

Ln 190: ensemble members. It is the first time here that ensemble members are mentioned, but the reader does not know here how these ensemble is developed.

We also mention "ensemble members" in the WRF model session, but we suggest removing "ensemble members" in this sentence, as it does not add information.

Figure 4: Can more be said about Figure 4. It now only gets 5 lines, which means maybe the figure is not needed. For example whether the data gaps occur particularly in LLJ favorable seasons or not, and whether that may result in a biased understanding of the LLJs.

We propose to add the following paragraph to clarify this point:

"Missing data at NS1 and BS1, particularly in LLJ-favorable seasons (spring), likely introduce a relative bias in annual LLJ statistics (the larger number of cases at BS2 relative to BS1 is illustrative of this). However, this has minimal impact on model evaluation since biases can be inferred from nearby FLSs, and the missing periods are removed from the modeled time series before evaluation."

Figure 4: the captions should explain what are NS1, NS2, BS1, BS2 and Osterild N.

We have clarified that these refer to the five sites.

Figure 5: I don't know whether it will appear later, but here as a reader I expect some words whether the climatology found is in agreement or disagreement with earlier studies over the North Sea and Baltic Sea. And if so, can you explain these differences.

We have added further comments and comparisons to previous findings, although not directly comparable due to spatial differences. We also offer plausible explanations for the observed differences between the two locations.

Figure 5: please add in the caption what is the time step of the data. Ie. whether we look at a frequency of the time in 10 min boxes or 1 h boxes.

Clarified that this is from hourly averaged data.

Figure 5: Please add whether the time is in UTC or local time in panel b.

We have now clarified that it is time-of-day in UTC time

Figure 5: Panel b shows a clearly different timing of the peak of the LLJ between the two seas. This needs to be mentioned and explained in the text.

See the comment above.

Ln 241: 2ms$^{-1}$ and 20% falloffs. Why do you deviate here from your more relaxed criterion explained in the Method (1.5 m/s)?

We decided, only for the purpose of selecting a number of days for the evaluation, that it would be best to select from a sample of more substantial cases rather than from a larger sample of weaker cases. When the data from these days are then analyzed, the more relaxed thresholds are used both for observations and models.

Figure 7: The error bar represents the spread among the five sites. Please be more specific whether the spread is 1 or 2 standard deviations, or is it really the min and the max. In the latter case it is better perhaps to turn into a whisker plot that also contains the quantiles.

We have clarified that these indicate 1 standard deviation between the five sites.

Figure 7d and 7e: the range is relatively small, so better start the y axis not at 0.

Fixed

Figure 8: there are more bars in the bar graphs than are in the legend. Please complete.

We have now added all of the labels explicitly.

Ln 279-280: If Fig 8 only gets one sentence of attention, is it then really needed?

We have moved Fig. 8 to the Appendix.

Ln 297: add a ) behind the "on average"

Fixed

Figure 10: caption "Maps of modelled LLJ occurrence …."

Fixed

Figure 12: in the WRF model climatology…. Please add in the caption which WRF ensemble member is used.

Good point. Fixed.

Table A1: header "Obuhkov length (L)" -> should be "Obukhov length ($L$)"

Fixed

**REVIEWER #2**

The manuscript focuses on low-level jets over the North Sea and the Baltic Sea. The authors discuss how these are simulated using the WRF model with various setups. They compare their results against observations as well as the NEWA and ERA5 datasets. The analysis is important not only from a scientific perspective but also for wind energy resource assessment and applications, although the latter appears somewhat inconsistent with the overall analysis presented in the paper. The manuscript is well-written and clear. However, it should be considered for publication only after a revision is undertaken.

Major Remarks:

1. The analysis appears to have been conducted in sections that seem somewhat disconnected from each other. It is unclear why the authors selected different

setups for the E and S simulations compared to the P simulations. Why not use the P domain with a 3 km resolution throughout to maintain consistency? While the one-way nesting approach and the 3 km resolution evaluation could support the argument that the findings are robust in the second setup, this inconsistency should be better justified. The same applies to the differences in the simulation periods for each run.

Thank you for raising this important point. The transition from the E to the P domain resulted from balancing two factors: managing computational costs for sensitivity experiments and expanding the spatial coverage to include the North and Baltic Seas in the climatology. The S experiments acted as an essential link, allowing us to transition from the best-performing ensemble member to a computationally efficient production setup. We have carefully evaluated the impacts of these differences and shared these observations in the paper. Although not optimal, we believe that the shift in the domain from experiments to climatology does not meaningfully diminish the relevance of either the sensitivity experiment or the findings in the five-year climatology. P_CLIM remains one of the best simulations on LLJ-related metrics. The second round of experiments needed different simulation periods because we were testing the sensitivities to running 24+12H runs vs. longer runs, which would mean sometimes having modeled LLJ results with longer lead times.

2. The authors should emphasize more on discussing how the model physics influence the outcomes of the experiments rather than focusing solely on the statistical evaluation.

We have added some further remarks on this with references to previous studies in the discussion.

3. The use of a adaptive timestep can result in unmonitored adjustments, potentially introducing differences in the model simulations and influencing model behavior, particularly under extreme conditions. The authors should provide a justification to ensure that the adaptive timestep does not affect the performance of each member.

All members used the same adaptive time-step settings, so any effects would be uniform. The key question is whether these unmonitored adjustments introduce spurious changes, leading to different LLJ evaluation outcomes due to minor variations

in settings or initial conditions. Initial testing showed no evidence of this; however, we lack fixed time-step control runs, primarily due to computational constraints.

4. Although there is a clear connection with wind energy resource assessment and applications, the section "Wind Energy Resources" does not align well with the rest of the manuscript and does not provide significant added value. Therefore, it should be removed.

Given suggestions by reviewer #1 and our own inclinations, we prefer to keep it in, especially given the relevance in serving as useful information in combination with existing wind atlases (GWA, NEWA, DOWA) for planning future wind farms.

Minor Remarks:

1. Line 108: Replace "E2" with "D2-E."

Thank you. Fixed.

2. Table 3: Timestep - Please indicate the exact setup of the adaptive timestep.

We have added the exact setup to the table

3. Figures 7, 8, etc.: Include the "light colors/dark colors" distinction in the legend.

We agree that it can be made more clear. We have updated it with a different, hopefully more clear, legend.